# Transposase-assisted target-site integration for efficient plant genome engineering

Peng Liu[1], Kaushik Panda[1], Seth A. Edwards[1,2], Ryan Swanson[1,2], Hochul Yi[3], Pratheek Pandesha[1,4], Yu-Hung Hung[1], Gerald Klaas[1], Xudong Ye[5], Megan V. Collins[6], Kaili N. Renken[6], Larry A. Gilbertson[5], Veena Veena[3], C. Nathan Hancock[6] & R. Keith Slotkin[1,2 ✉]

The current technologies to place new DNA into specific locations in plant genomes are low frequency and error-prone, and this inefficiency hampers genome-editing approaches to develop improved crops[1,2]. Often considered to be genome 'parasites', transposable elements (TEs) evolved to insert their DNA seamlessly into genomes[3–5]. Eukaryotic TEs select their site of insertion based on preferences for chromatin contexts, which differ for each TE type[6–9]. Here we developed a genome engineering tool that controls the TE insertion site and cargo delivered, taking advantage of the natural ability of the TE to precisely excise and insert into the genome. Inspired by CRISPR-associated transposases that target transposition in a programmable manner in bacteria[10–12], we fused the rice *Pong* transposase protein to the Cas9 or Cas12a programmable nucleases. We demonstrated sequence-specific targeted insertion (guided by the CRISPR gRNA) of enhancer elements, an open reading frame and a gene expression cassette into the genome of the model plant *Arabidopsis*. We then translated this system into soybean—a major global crop in need of targeted insertion technology. We have engineered a TE 'parasite' into a usable and accessible toolkit that enables the sequence-specific targeting of custom DNA into plant genomes.

Transgenesis in plants is widely used to both generate research tools and engineer crop improvement[13]. Traditional transgene integration occurs at random locations in the plant genome[14], which can generate unintended mutations and subject the expression of the transgene to undesired position effects[15,16]. Sequence-specific target-site integration of transgenes is highly desired and has been attempted in plants for over a decade (reviewed previously[1]). Although there has been progress[17,18], the low efficiency and quality of current target-site integration technologies hampers crop genome engineering[2].

Several approaches have been taken in plants to insert transgenic sequences into specific genomic locations[17–19]. Homologous recombination (HR) in plants has been demonstrated, but the rate is extremely low and it is therefore not used[20]. Programmable nuclease systems that create sequence-specific DNA breaks, such as CRISPR–Cas9, have improved targeted integration[1]. Prime editing techniques in plants enable the insertion of small sequences up to 34 base pairs (bp) at a targeted site[21], but larger insertions are required to encode new extrinsic traits[22]. Homology-directed repair (HDR) can integrate new DNA through the resolution of induced double-stranded DNA breaks by recombination with a supplied DNA sequence that matches both flanks of the cleaved DNA[23]. HDR occurs at a low frequency in plants as the repair of the DNA breaks occurs primarily by non-homologous end joining (NHEJ), which can include the integration of random fragments of DNA[24]. The preference for NHEJ repair in plants has been taken advantage of to knock-in supplied extrachromosomal DNA during the repair of a CRISPR-induced double-stranded break[25,26]. HDR and NHEJ knock-ins are frequently subject to deletions of the flanking target-site DNA and/or the sequence that was delivered (the cargo)[17,27]. Owing to the low integration frequency and high deletion rate, these approaches involve the expensive production of many transgenic plants followed by laborious screening for a rare intact targeted integration event.

TEs, and more specifically class II DNA transposons, can 'cut and paste' their DNA into new genomic locations. TE-encoded transposase proteins excise their corresponding TE DNA from a donor position, protect this extrachromosomal DNA from nucleases[3,4] and accurately insert the TE DNA into the genome at a new integration site[3–5] (Extended Data Fig. 1). The selection of the integration site is highly variable among different TE types, and can target accessible chromatin[6], a favoured chromatin state[7], sites of double-stranded breaks[8] or a short sequence such as TAA[9]. Bacterial genomes contain natural CRISPR-associated transposases that have been engineered using different CRISPR guide RNAs (gRNAs) to transpose the TE to specific sites in bacterial genomes[10–12]. These systems lack Cas cleavage activity and are dependent on the transposase to perform integration. Moreover, other DNA transposons have been synthetically combined with either an active[28] or catalytically inactive[29,30] programmable nuclease to target TE transposition in animal cell and tissue cultures. Here we generated a transposase-assisted

[1]Donald Danforth Plant Science Center, St Louis, MO, USA. [2]Division of Biological Sciences, University of Missouri, Columbia, MO, USA. [3]Plant Transformation Facility, Donald Danforth Plant Science Center, St Louis, MO, USA. [4]Division of Biology and Biomedical Sciences, Washington University, St Louis, MO, USA. [5]Bayer Crop Science, St Louis, MO, USA. [6]University of South Carolina-Aiken, Aiken, SC, USA. ✉e-mail: kslotkin@danforthcenter.org

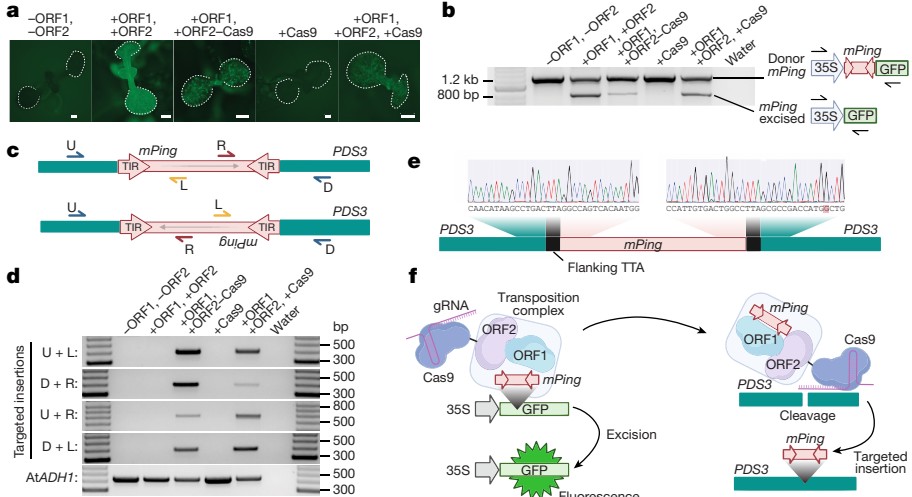

**Fig. 1 | The combined activities of a transposase and programmable nuclease result in targeted insertion. a**, Excision of the *mPing* TE from a GFP reporter restores fluorescence. *Arabidopsis* seedlings were imaged; the cotyledons are outlined with a white dashed line. 'ORF2–Cas9' represents a translational fusion of these proteins. Scale bars, 500 μm. **b**, Excision of *mPing* assayed by PCR in pooled seedlings. The top band represents *mPing* within GFP (donor position), and the smaller band is generated after *mPing* excision. **c**, PCR primer design for detecting targeted insertions of *mPing* at the *PDS3* locus. U and D are *PDS3* primers that surround the CRISPR target site. R and L are *mPing* primers. TIR, terminal inverted repeat. **d**, PCR amplification analysis of targeted insertions of

*mPing* at the *PDS3* locus in pooled seedlings. At*ADH1* was the PCR control. **e**, Sanger sequencing of the insertion junctions generated after *mPing* insertion into *PDS3*. The light grey bars behind the DNA-sequencing peaks represent quality scores for each base call. Bases highlighted in red are mismatches compared with the reference sequence. The flanking TTA sequence that comes with *mPing* from the donor site is annotated. **f**, Model of targeted insertion of *mPing* at the *PDS3* locus. A functional ORF2–Cas9 fusion protein excises *mPing* out of the 35S–GFP donor site, cuts the *PDS3* gene guided by the gRNA and *mPing* is inserted into the *PDS3* target site. The diagram in **f** was created using BioRender.

target-site integration (TATSI) system by co-expressing catalytically active programmable nucleases with the rice *Pong* DNA transposon system[31]. We accomplished transposase-mediated targeted insertion in a model plant (*Arabidopsis*) and translated this technology into a crop plant (soybean). We demonstrate a higher frequency and accuracy of targeted integration compared with currently used methodologies, as well as the delivery of enhancers and gene cargos, in individual plants rather than cultured cells.

## Combining a transposase and Cas protein

We aimed to combine the function of CRISPR–Cas targeting and TE transposition into a single system by fusing the rice *Pong* transposase proteins with programmable nucleases such as Cas9. We began with a previously established *Arabidopsis* vector system in which hyperactive versions of the rice *Pong* proteins (named ORF1 and ORF2) have been removed from the TE and are ubiquitously expressed by genic promoters. Both *Pong* ORF1 and ORF2 are necessary for the excision and insertion of the non-autonomous rice TE *mPing*[9] (Extended Data Fig. 1). The Pong ORF1 protein binds to 15 bp terminal inverted repeats on the ends of the 430 bp *mPing* element, which itself is flanked by TTA or TAA repeats that are necessary for excision[32] (Extended Data Fig. 1). We synthetically fused Cas9, the Cas9(D10A) nickase (cleaves only one strand of DNA) or dCas9 (catalytically dead version) to the N- and C-terminal ends of *Pong* ORF1 and ORF2, resulting in 12 fusion protein transgene configurations (Extended Data Fig. 2a). We used a two-step transformation strategy, in which *mPing* is located within a GFP expression cassette that was previously integrated into the *Arabidopsis* genome (donor element)[9]. This line was then germinally transformed with our *Pong-Cas9* fusion proteins and tested for the excision of *mPing*. This includes a transgene version in which ORF1, ORF2 and Cas9 are all present but unfused. *mPing* excision, demonstrated by GFP fluorescence and verified by PCR, was detected for all 12 fusion proteins and controls in which ORF1 and ORF2 are present with or without Cas9 (Fig. 1a,b and Extended Data Fig. 2b–d), demonstrating

that ORF1 and ORF2 are capable of transposase function with fusions on either end.

Each transgene used a CRISPR gRNA that targets the *Arabidopsis PHYTOENE DESATURASE3 (PDS3)* gene. We verified that that our CRISPR–Cas9 system was functional through a mutation-detection assay at *PDS3* (Extended Data Fig. 3a) and the observation of *pds3*-mutant white seedlings and plant sectors with the catalytically active Cas9 protein (Extended Data Fig. 3b,c). We observed a decrease in the rate of mutations induced by Cas9 when fused to ORF1 or ORF2, and determined that the N-terminal fusions to Cas9 have higher activity than C-terminal fusions (Extended Data Fig. 3c).

We used a combination of four PCR reactions to assay for targeted insertion of *mPing* at the *PDS3* gRNA target site (Fig. 1c). Of the 12 tested protein fusions and 4 controls, 2 provided reproducible and high-rate targeted insertion of *mPing* (Fig. 1d and Extended Data Fig. 4a). These two are the fusion of Cas9 to the C terminus of ORF2, and unfused Cas9 (Fig. 1d). Targeted insertion occurred only in plants that possessed both a fully catalytically active Cas9 and the ORF1 + ORF2 proteins. We verified that the ORF2–Cas9 fusion protein was intact and not generating separate versions of the two proteins (Extended Data Fig. 4b). Targeted insertions were verified by Sanger sequencing of the insertion junctions (Fig. 1e and Extended Data Fig. 4c).

To determine whether *mPing* targeted insertion occurs specifically with the Cas9 nuclease, or whether other programmable nucleases could be used, we combined *Pong* ORF1 and ORF2 with *Lachnospiraceae bacterium* Lb*Cas12a*, a distinct CRISPR-guided nuclease[33]. Three configurations of Cas12a were tested for *mPing* targeted insertion: Cas12a–ORF2, ORF2–Cas12a and unfused Cas12a. As with Cas9, the ORF2–Cas12a fusion protein and unfused Cas12a resulted in *mPing* excision and targeted insertion (Extended Data Fig. 5), demonstrating that a range of nucleases can be used in combination with *Pong* transposase proteins to perform targeted integration. These findings demonstrate that, after excision from the donor site, the *mPing* TE inserts into double-stranded breaks generated by a programmable nuclease system (Fig. 1f).

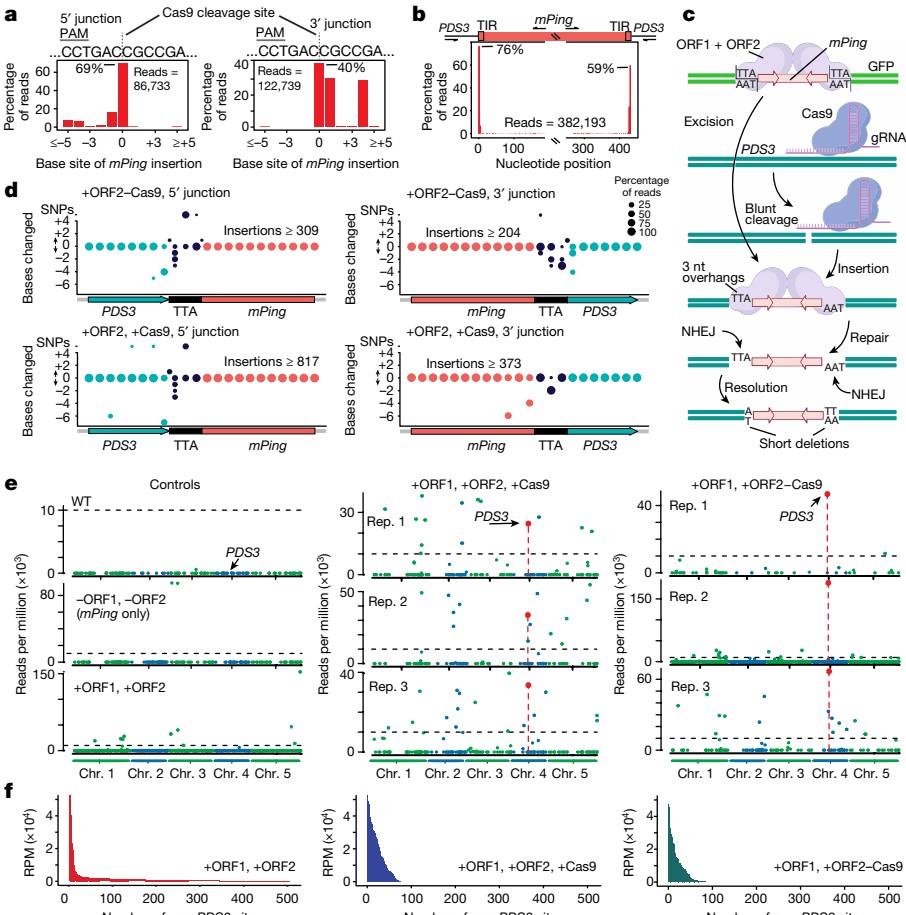

**Fig. 2 | Precision of targeted insertion events. a**, The dashed line marks the Cas9 cleavage site on the *PDS3* target sequence before TE integration. Insertion sites are assayed at the 5′ (relative to *PDS3*) (left) or 3′ junction (right) of *mPing* insertions. The '0' site marks insertion at the exact Cas9-cleavage site. PAM, protospacer adjacent motif. **b**, Sequencing analysis of targeted insertion junction points mapped to *mPing* indicates how much of the *mPing* element was delivered to the targeted insertion site. The *x* axis shows the nucleotide position along the *mPing* element. The break in the *x* axis represents the interior of *mPing* that was not assayed. **c**, Model *mPing* excision by an ORF2 transposase-generated staggered break, blunt cleavage of the target site by Cas9, then integration, repair and resolution of *mPing* at the target site by NHEJ. The diagram was created using BioRender. **d**, Nucleotide (nt) variation at the junction of *mPing* insertions into *PDS3*. The precision of each nucleotide at the insertion site was determined on the 5′ junction (left) or 3′ junction (right). The size of the circle represents the percentage of reads in which that nucleotide is as expected (*y* = 0), has an insertion (*y* ≥ 1) or deletion (*y* ≤ −1). The number of SNPs at the insertion site is shown at the top of the *y* axis. Pearson's $\chi^2$ tests were used to test the statistical significance of the difference in polymorphism between the two protein configurations. **e**, *mPing* insertion sites in pooled seedlings. The *Arabidopsis* nuclear genome is displayed on the *x* axis. The *PDS3* target site is shown with an arrow and red datapoint. The scale of each *y* axis was determined by the maximum datapoint. A dashed line at 10,000 reads per million (RPM) is shown for each sample. Chr., chromosome; Rep., distinct biological replicates; WT, wild type. **f**, Quantitative analysis of the number and read support of free-transposition sites in pooled replicates for each genotype.

## Precision at targeted insertion sites

To determine the precision of junction sequences between *mPing* and the *PDS3* target site, we performed amplicon deep sequencing of pooled *PDS3* insertion events. We analysed more than 1,703 distinct targeted insertion junctions at the *PDS3* locus and found that the majority of the insertions occur at the CRISPR cleavage site or within 4 bp (Fig. 2a). Second, the majority (66.8%) of *mPing* inserted elements have intact ends and are full length (over the region assayed) (Fig. 2b), demonstrating that, when *mPing* inserts, the complete or near-complete element is delivered. This full or near-full insertion of the cargo TE is probably due to the binding of the TE by the transposase proteins while extrachromosomal[5], protecting the DNA ends from nucleases and deletions (Fig. 2c). Third, a three-nucleotide target-site duplication with the sequence TTA/TAA is a feature produced after *mPing* free transposition[31] (Extended Data Fig. 1). The TTA/TAA sequence does not need to be present at the target site for TATSI targeted integration, but a new flanking TTA/TAA sequence is often observed after TE integration at

the TATSI target site (Extended Data Fig. 4c). This new TTA/TAA flanking sequence is generated from the donor site after a staggered cut by the *Pong* transposase. The presence of the flanking TTA/TAA bases demonstrates that the targeted insertion event only occurs after (1) ORF1 and ORF2 excise *mPing* out of the donor site, and (2) cleavage of the target site by the programmable nuclease (Fig. 2c).

We observed examples of both perfectly accurate insertions of *mPing* at the target site as well as sequence polymorphism generated upon insertion. The majority of these polymorphisms were located at the junction nucleotides between the flanking TTA/TAA bases and target-site DNA (Fig. 2d). A 1 bp single-nucleotide polymorphism (SNP) or insertion was frequently observed on either the 5′ or 3′ end of the insertions (relative to the *PDS3* gene direction), while deletions up to 7 bp can occur in the flanking TTA/TAA bases or into the *PDS3* gene (Fig. 2d). The most common variation is a short 1–3 bp deletion at the junction between the flanking TTA/TAA bases and *PDS3* (Fig. 2d and Extended Data Fig. 4c). We compared the level of polymorphism generated at the target site between the fused and unfused configurations

and found that the difference was statistically significant ($P < 0.01$), with the unfused configuration generating marginally more 'base for base' perfect insertions compared with the fused configuration (Fig. 2d). Together, these data demonstrate that the *mPing* cargo is intact after targeted insertion, with small deletions occurring at the bases at the junction of *mPing* and the target-site DNA. These small deletions are probably caused by NHEJ during the repair of the integration junction that is necessary to resolve the *mPing* extrachromosomal DNA that has three-nucleotide overhangs with the blunt ends of the Cas9 nuclease-cleaved insertion site (Fig. 2c).

## Off-target rate

To investigate whether *mPing* insertions occur at other regions of the genome besides the CRISPR–Cas on-target site, we performed insertion-seq to identify all of the rice *mPing* insertion sites in the *Arabidopsis* genome. We captured the sequences flanking the *mPing* terminal inverted repeats by generating a genomic library in a plasmid vector, then performing PCR between a vector primer and an *mPing* internal primer, followed by deep sequencing (Methods). This technology is more sensitive for the detection of insertions that may occur in only one or few cells compared to commonly used whole-genome sequencing. We used three control lines, including wild-type (no ORF1, ORF2 or CRISPR–Cas system), a line with only *mPing* (without ORF1/ORF2 and Cas9) and two biological replicates of pooled seedlings that have an active *mPing* transposition system (+ORF1, +ORF2) but lack the CRISPR–Cas system. The +ORF1, +ORF2 samples represent unfettered free transposition of *mPing* into the *Arabidopsis* genome (a qualitative analysis is shown in Fig. 2e (left)), and have hundreds of *mPing* insertion sites throughout the genome (a quantitative analysis is shown in Fig. 2f (left) and Extended Data Fig. 6a). When the CRISPR–Cas system is added in +ORF1, +ORF2, +Cas9 (unfused) plants, the number of *mPing* insertion sites is reduced in this pool of seedlings (Fig. 2f (middle) and Extended Data Fig. 6a), and the targeted site of *PDS3* represents one of the sites of *mPing* insertion (Fig. 2e (middle)). In fused +ORF1, +ORF2–Cas9 pooled seedlings, the primary location of *mPing* insertion is the targeted *PDS3* site (Fig. 2e (right)), although other sites of insertion can also be detected (Fig. 2f (right) and Extended Data Fig. 6a). The non-*PDS3* *mPing* insertion sites are not directed by CRISPR–Cas9 cleavage, as none of the off-target sites are in common between replicates with Cas9 (Extended Data Fig. 6a), and *mPing* does not insert at predicted gRNA off-target sites (Extended Data Fig. 6b). If we use the approach of recent publications[22] that interrogate only off-target insertions at sites that partially match the gRNA sequence, we find that TATSI has an off-target rate as low as other state-of-the-art target-site integration technologies in plants (Extended Data Fig. 6b). These data demonstrate that the *mPing* insertions that are at other locations besides *PDS3* represent free-transposition sites generated by the active *mPing* transposition system and not off-target sites of CRISPR–Cas9 cleavage. We observe that the number of free-transposition sites decreases in either fused or unfused lines with the CRISPR–Cas system (Fig. 2f and Extended Data Fig. 6a), suggesting that *mPing* is channelled to a targeted location by the double-stranded break generated by CRISPR–Cas.

## Programmability of targeted insertions

We demonstrated the programmability of TATSI by changing the CRISPR gRNA and targeting *mPing* insertion to either an exon of the *ALCOHOL DEHYDROGENASE1* (*ADH1*) gene or to the non-coding region upstream of *ACTIN8* (*ACT8*). These gRNAs were verified to be functional on the basis of a Cas9 mutation-detection assay (Extended Data Fig. 3a). Targeted insertions of *mPing* at these loci were detected by PCR and verified by Sanger sequencing of the insertion junctions (Fig. 3a). We also performed multiplexed targeted insertion by generating two gRNAs (targeting *ADH1* and *ACT8*) from a single transcript (Extended Data

Fig. 5a). We detected *mPing* at both the *ADH1* and *ACT8* loci (Extended Data Fig. 5c–h), demonstrating that *mPing* can be distributed to multiple targeted loci from one transgenesis event.

As TEs are the targets of DNA methylation in plants[34] and, through transposition, they can recruit methylation to new loci[35], we tested whether *mPing* brings DNA methylation to an unmethylated *ADH1* exon after TATSI insertion. After targeted insertion into *ADH1*, both *mPing* and the flanking *ADH1* exon are not methylated above the background level found at this locus before *mPing* insertion (Extended Data Fig. 7a). The lack of DNA methylation attracted by *mPing* is probably due to it being an unexpressed non-autonomous TE and the fact that it is a foreign TE that is not recognized by identity-based silencing in *Arabidopsis*[36].

## Rate of targeted insertion

The above targeted insertions were tested using a two-component transformation strategy in which the donor *mPing* was previously integrated into the genome on a separate transgene from the ORF1/ORF2/Cas9/gRNA transgene. We generated a one-component system in which the *mPing* donor site was present on the same transgene (Extended Data Fig. 7b). Similar to the two-component system, this one-component system successfully produced targeted insertions of *mPing* (Extended Data Fig. 7c,d), simplifying the transgenesis required for targeted insertion. We used this one-component system and the non-deleterious intergenic target site upstream of *ACT8* to test the rate of targeted insertion in *Arabidopsis* plants. In 120 individual first-generation (T1) transgenic plants with the ORF2–Cas9 fusion, we detected 75.0% with *mPing* excision and 6.7% with targeted insertion (Fig. 3b,c). We validated that these plants have both sides of the *mPing* element at the targeted insertion site by PCR. In comparison, the unfused +ORF2, +Cas9 configuration generated higher rates for both *mPing* excision (98.7%) and targeted insertion (35.5%) (Fig. 3b,c). Our observed rate of targeted insertion is an improvement on the reported rates of T1 site-specific integration compared with HR (0.24%)[37], HDR (0.68–2.4%)[38,39] or NHEJ knock-in (4.8%)[40] in *Arabidopsis*. Only plants that displayed excision had the potential for targeted insertion (Fig. 3b) and, of the plants that have excision, 36–45% have targeted insertion, suggesting a limiting step for targeted insertion is *mPing* excision. Furthermore, we performed a knock-in strategy similar to homology-independent targeted integration (HITI) in which CRISPR–Cas9 both excises the cargo from the donor site and creates the break at the target site[41]. We found rates of targeted insertion for HITI intermediate to the fused and unfused versions of TATSI (Fig. 3c), suggesting that the higher rate of targeted insertion in the unfused configuration of TATSI may be due to transposase binding of the extrachromosomal *mPing* element.

## Insertion of enhancer and gene cargos

To test the cargo capacity that can be delivered by *mPing*, we engineered four distinct variations that add additional sequences to the rice 430 bp *mPing* element. The first adds a synthetic array of enhancers composed of six heat shock elements (HSEs) (Fig. 3d and Extended Data Fig. 8a). This synthetic 444 bp *mPing_HSE* element is capable of excision (Fig. 3c and Extended Data Fig. 8b) and targeted insertion into the region upstream of the *ACT8* gene (Fig. 3c and Extended Data Fig. 8c,d). Sanger sequencing verified that all six HSEs were delivered to the targeted insertion site (Extended Data Fig. 8e).

We next tested larger cargos by embedding the protein-coding region of the herbicide bialaphos resistance gene (*bar*) into *mPing*, generating the 1,002 bp synthetic element *mPing_bar_CDS* (Fig. 3d). We also embedded the *bar* gene including the NOS promoter and terminator into *mPing*, creating the 1,563 bp *mPing_bar* element (Fig. 3d). The agricultural use of herbicide-resistant plants has reduced cost and increased yields[42], and targeting the insertion of these resistance genes

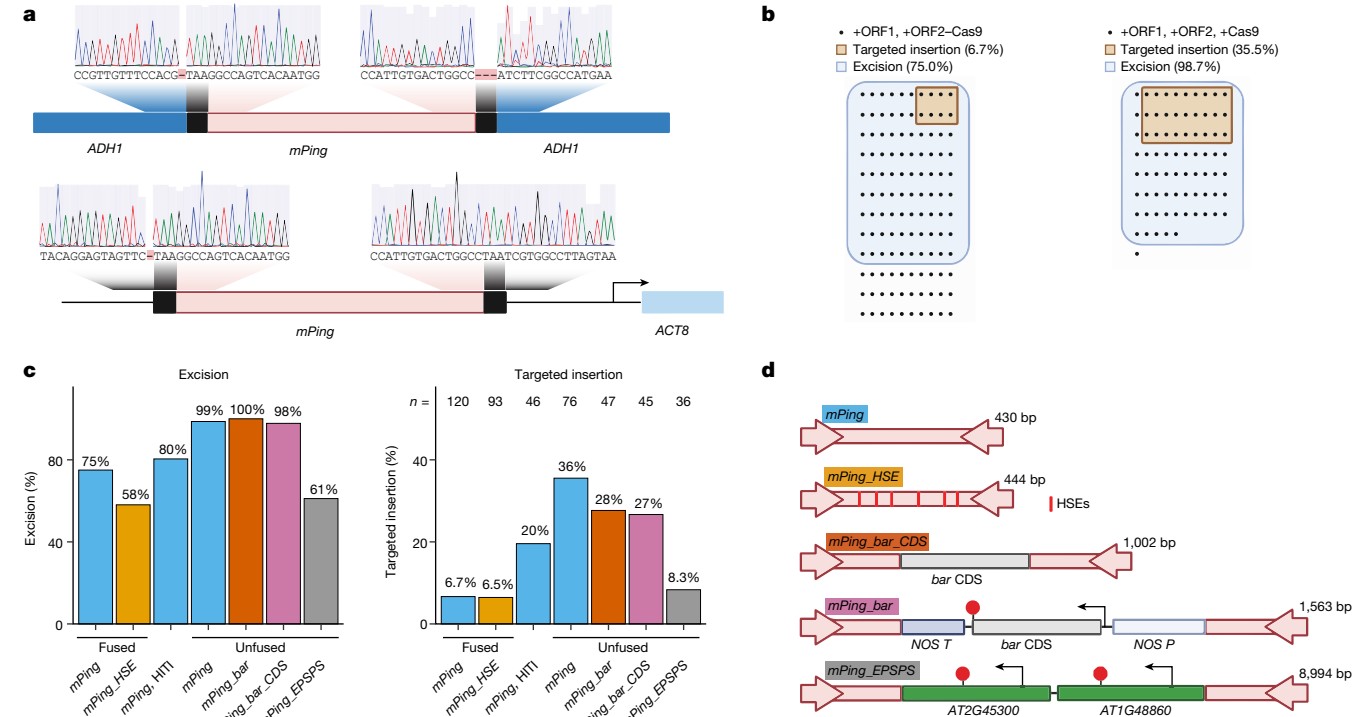

**Fig. 3 | Programmability of the insertion site and cargo. a**, Sanger sequencing analysis of the junctions of *mPing*-targeted insertion events in the *ADH1* gene and in the non-coding region upstream of *ACT8*. **b**, Visualization of the rate of targeted insertion upstream of *ACT8*. Each dot represents a distinct T1 transgenic plant, and the plants with *mPing* excision (blue) and targeted insertion (orange) are marked. **c**, Measurement of the excision frequency of *mPing* from the donor site (left) and rate of targeted insertion (right). *n* is the number of T1 transgenic *Arabidopsis* plants analysed. The colour of each data bar corresponds to the *mPing* cargo colour code in **d. d**, The cargo of different *mPing* versions demonstrated to excise and undergo targeted insertion in the *Arabidopsis* genome. *NOS P–bar–NOS T* is an expression cassette that expresses a herbicide-resistance gene, and *bar* CDS is the protein-coding region without the promoter and terminator. *mPing* versions are not drawn to scale, and the size of each is indicated.

avoids the creation of new mutations and allows for trait stacking at closely linked positions[43]. Both the *mPing_bar_CDS* and *mPing_bar* elements were capable of excision (Fig. 3c and Extended Data Fig. 9a) and targeted insertion into the non-coding region upstream of *ACT8* (Fig. 3c and Extended Data Fig. 9b,c) in 26.7% and 27.7% of T1 *Arabidopsis* plants, respectively (Fig. 3c). We confirmed using Sanger sequencing that the entire cargo (*bar* expression cassette in *mPing_bar* and CDS region in *mPing_bar_CDS*) were delivered intact and mutation-free to the targeted insertion site (Extended Data Fig. 10 and 11). We further tested the size limit of cargo delivery and found the TATSI system can deliver cargo of two *Arabidopsis* endogenous genes totalling 8.6 kb (creating the 8,994 bp *mPing_EPSPS*; Fig. 3c,d). The frequency of targeted insertion was reduced to 8.3% with this large cargo (Fig. 3c), suggesting that smaller cargos are more efficient for targeted insertion using TATSI.

## Targeted insertions in soybean

To demonstrate the commercial use of TATSI in a crop plant, a series of *mPing* vectors was transformed into soybean (*Glycine max* var. Williams 82). Soybean has the estimated fourth largest global harvest and is a critical global source of protein and oil[44]. Soybean has been a target of crop improvement, with 94% of the US soybean crop now transgenic[45]; however, considerable agricultural improvement remains possible for soybeans[46]. Target-site integration of transgenes in the soybean genome remains inefficient, with new approaches at multinational seed companies reaching rates of only 3.4% (ref. 47). Previous reports demonstrated the free transposition of the rice *mPing* element by ORF1 and ORF2 when transformed into soybean[48]. To determine whether *mPing* can be used for improved target-site integration in

soybean, we tested seven TATSI transgene configurations (Fig. 4a), and each of these was co-expressed with a gRNA targeting the intergenic region *DD20*, a safe-harbour location to target transgene integration.

Both ORF2 + Cas9 fused and unfused configurations functioned to generate high rates of *mPing* targeted insertion in soybean plants (Fig. 4b). The same ORF2–Cas9 fusion with a 1×G4S linker that was functional in *Arabidopsis* (Fig. 1b–d) did not have Cas9 activity in soybean (Fig. 4b (light blue)), whereas a longer 3×G4S linker generated both the *mPing* excision (76.2%) and Cas9 activity (38.1%) required for TATSI targeted insertion (15.9%) (Fig. 4b (dark blue)). These data are consistent with *mPing* excision rates in yeast, in which the longer 3×G4S linker displayed higher activity than the 1×G4S linker (Extended Data Fig. 3d). Similar to *Arabidopsis*, the ORF2 + Cas9 unfused transgene configuration in soybean (1) has higher rates of *mPing* excision (92.7%) and targeted insertion (18.2%) (Fig. 4b (yellow)); and (2) the insertion junctions also have small flanking deletions. One *mPing* insertion at the *DD20* site has a fully accurate junction on the 3′ side and a three-nucleotide deletion of the flanking TTA sequence on the 5′ side (Fig. 4c). (3) Insertion sequencing (insertion-seq) of an R0 regenerated soybean plant with the ORF2 + Cas9 unfused transgene indicated seven major *mPing* insertion sites in this plant, one of which is the targeted insertion at the *DD20* location (Fig. 4d (red)). The other sites of *mPing* insertion do not have sequence similarity to the gRNA (Fig. 4d) and, rather, are at sites of the known TTA/TAA preference sequence for *mPing* insertion[9], thus representing free *mPing* transpositions.

The delivery of enhancer elements and a gene cassette was also tested in soybean by replacing *mPing* with *mPing_HSE* or *mPing_bar* in the unfused +ORF2, +Cas9 transgene configuration (Fig. 4a (orange and red)). Targeted insertion of *mPing_HSE* occurs with similar flanking NHEJ deletions as seen in *Arabidopsis* (Fig. 4c), while the rate of targeted

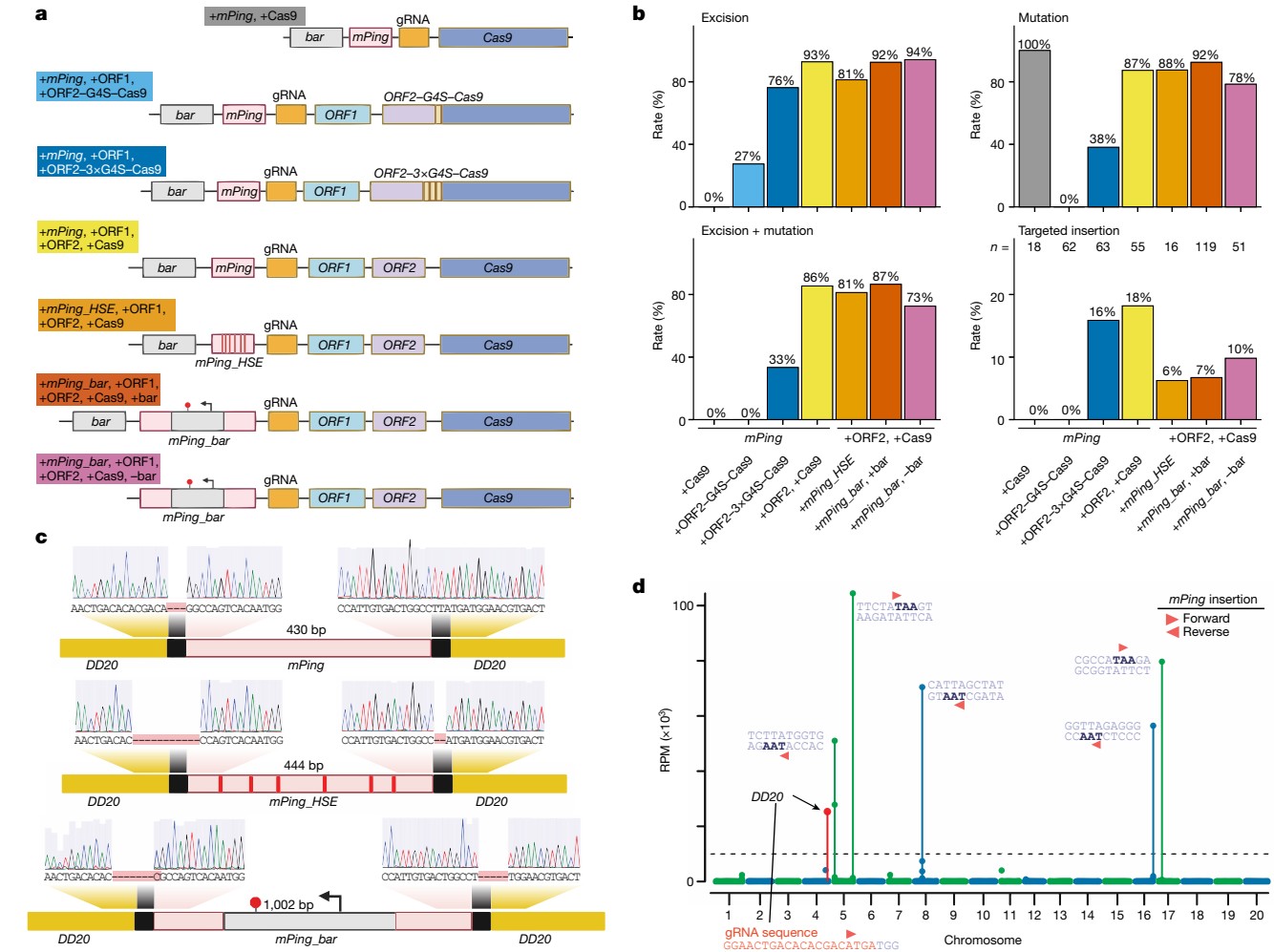

**Fig. 4 | Targeted insertions in the soybean genome. a**, The configurations of seven different transgenes that were tested for targeted insertion. The label colour corresponds to the data bars in **b**. **b**, Rates of *mPing* excision (top left), Cas9-mediated mutations (SNPs) (top right), plants with both excision and mutation (bottom left) and targeted insertions (bottom right) in transgenic regenerated (R0) soybean plants. Both junctions of *mPing* must be found at the *DD20*-targeted insertion site to be counted as a positive targeted insertion event. *n* is the number of transgenic individuals tested. **c**, Sanger sequencing of the junctions of *mPing*, *mPing_HSE* and *mPing_bar* insertions into the soybean *DD20* non-coding target site. **d**, Insertion-seq defines the locations of *mPing* in the genome of a single R0 soybean plant. The soybean nuclear genome is displayed on the *x* axis. Insertion is detected at the *DD20* targeted insertion site (red datapoint) as well as at six other sites. These other sites do not have similarity to the gRNA sequence, but are TAA/TTA sites favoured for insertion from free transposition of *mPing*. The triangles denote the orientation of *mPing* insertion. A black dashed line at 10,000 RPM is shown.

insertion is reduced to 6.3% (Fig. 4b). The 1.5 kb *mPing_bar* element was delivered at a rate of 6.7% (Fig. 4b). This experiment confirms the targeted delivery of custom DNA into the soybean genome using TATSI, while also suggesting that there is an efficiency penalty for altering the *mPing* sequence.

To test whether *mPing_bar* was functional in soybean, we constructed a transgene in which the only herbicide-resistance gene in the vector was present within *mPing* (Fig. 4a (purple)). In soybean plants, this *mPing_bar* element confers herbicide resistance, enabling the recovery of transgenic cells after transformation and growth on medium with herbicide. Similar to other *mPing* cargos, this *mPing_bar* element undergoes excision (94.1%) and targeted insertion (9.8%) (Fig. 4b). We also identified regenerated soybean plants with *mPing_bar* at the *DD20* insertion site but that lack the *mPing_bar* element at the parent transgene donor site (Extended Data Fig. 12a,b). One plant has a partial parent transgene integration without the donor *mPing_bar* element (Extended Data Fig. 12b (plant 2)), while a different plant has no parent transgene (Extended Data Fig. 12b (plant 3)). Both plants with targeted *mPing_bar* insertion are herbicide resistant, and therefore the

*mPing_bar* element is capable of functionally driving this trait when mobilized away from the parent transgene.

## Discussion

Prokaryotic CRISPR-associated transposases and similar synthetic systems have been demonstrated in bacterial cultures, animal cell culture and tissue cultures[12,28]. Here we produced a functional TATSI system to generate targeted insertions in whole individuals of the model plant *Arabidopsis*, and translated this technology into soybean plants, which represent a critically important crop for global oil and protein production. Two configurations were successful for targeted insertion, with the ORF2 transposase protein either translationally fused or unfused to the programmable nuclease. The fusion decreases transposase activity but also increases the ratio of on-target insertions. TATSI has the potential to work in any transformable crop genome, with the exception of rice, in which it is likely to be epigenetically silenced. The rice *mPing/Pong* system avoids identity-based silencing in *Arabidopsis* and soybean because it is foreign to these genomes, and it avoids expression-based

silencing because *mPing* is not expressed and ORF1/ORF2 are expressed in *trans* by genic promoters and terminators that drive TE activity[36]. For rice, a TE from a different genome could be engineered for TATSI.

The critical factor for any targeted integration tool is the frequency of on-target site insertion. TATSI represents a fold-change improvement over other methods of targeted insertion in the soybean genome. TATSI offers an improvement in *Arabidopsis* compared with HR, HDR or NHEJ knock-in. These other technologies are the product of years of refinement, and we expect that TATSI will further improve in the coming years. Data suggest that a limiting factor for targeted insertion is the rate of TE excision. Thus, the use of hyperactive versions of *mPing* such as *mmPing20*[49] may quickly increase the rate of targeted insertion. Moreover, all of the necessary components for targeted integration can be packaged into an all-in-one transgene for one-step delivery, and we identified a case in which the targeted integration is present in a R0 regenerated soybean plant without the parent transgene (Extended Data Fig. 12), skipping the requirement to segregate away the machinery required for targeted insertion.

Transpose proteins continue to bind to TE ends after excision from the donor site, protecting the free ends of the TEs while they are extrachromosomal[3,4]. We find that, after targeted insertion, the delivered TE is most often complete and rarely mutated, probably because the transposase proteins bind to *mPing* ends and protect these regions from nucleases while the DNA is extrachromosomal. We speculate that this is why TATSI in the unfused configuration results in a higher insertion rate compared with HITI, in which the cargo is unbound and unprotected while extrachromosomal. Our results suggest that *mPing* insertion is primarily mediated by NHEJ and not by transposase proteins[41]. We found that single-nucleotide insertions and small deletions are common at both the flanking TTA/TAA bases and the flanking insertion-site DNA. These small deletions probably represent degradation of the cut site after nuclease cleavage but before *mPing* insertion and repair of the junctions. The TTA/TAA bases flanking the TE are necessary for *mPing* excision[32] and, as one or both of these sites is often mutated after insertion mediated by TATSI, this results in targeted insertions being unable to excise out from their targeted integration sites. By contrast, insertions generated by free transposition will have an intact target-site duplication (TTA sequences at both ends) and will be capable of excision out of their insertion site.

Drawbacks to TATSI that can be improved in the future include controlling the orientation of *mPing* insertion (which is currently uncontrolled) and reducing free transpositions. Off-target insertions occur with both prokaryotic CRISPR-associated transposases and similar eukaryotic synthetic systems[12,28,29]. In TATSI, these untargeted insertions are the product of the active TE, and not off-target cleavage by the catalytically active CRISPR–Cas9. Free transposition insertions of *mPing* are reduced by the presence of Cas9, suggesting that the insertion of *mPing* is being directed to the on-target site by the presence of the Cas9-induced double-stranded DNA break. Off-target insertions during targeted integration are generally tolerated in crop production to a much higher extent compared with in medical or therapeutic applications. First, our method of detecting off-target effects is much more sensitive than other reports. Second, during transgenic crop production, the transgene is inserted into the genome of a transformation variety, and then introgressed (repeatedly back-crossed) into the ever-changing elite germplasm before deployment for agriculture[50]. This introgression process segregates away the rest of the genome and places the insertion event into a new genetic background, effectively removing any free-transposition events that are not tightly linked.

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

# Methods

## Transgene production

Transgenes were generated using the oligos listed in Supplementary Table 1. First, *Pong* ORF1 and ORF2, including promoters and terminators (sequence information is from pWMD_23)[51], were synthesized by GeneArt (Thermo Fisher Scientific) and cloned into a pHEE401E vector through In-Fusion cloning (Takara). Second, the Cas9, Cas9(D10A) nickase and dCas9 derived from *S. pyogenes*[52–54] were amplified from pHEE401E, pDe-Cas9-D10A and pDIRECT_21B, respectively. Third, a G4S flexible linker was added through PCR and each Cas9 version was fused to ORF1 or ORF2 by In-Fusion reactions. Fourth, a gRNA cassette was created and inserted through a GoldenGate reaction. Each transgene used a CRISPR gRNA that was previously demonstrated to cleave within the *Arabidopsis PDS3* gene[55], *ALCOHOL DEHYDROGENASE1* (*ADH1*) or upstream of the *ACTIN8* gene (*ACT8*) (Supplementary Table 1). For the one-component system, the *mPing* donor was amplified from genomic DNA of *Arabidopsis* line with a previously integrated *mPing* element[9], and cloned into the ORF2-Cas9 fusion vector described above. All *mPing* elements and derivatives have flanking TTA sites engineered into the donor site.

To test *Lb*Cas12a, DNA fragments containing multiplexed gRNA sequences were synthesized by Genewiz and cloned into the vector that expresses *Pong* ORF1 and ORF2 by an In-Fusion reaction. The multiplexed targeting strategy includes two gRNAs (targeting *ADH1* and *ACT8*) from a single transcript (as described previously[56]). DNA fragments of *Cas12a* were synthesized by IDT and added by In-Fusion reactions. The *Cas12a* sequence from *Lachnospiraceae* bacterium was optimized and provided by Bayer Crop Science.

To test the delivery of HSEs, *mPing*_HSE was synthesized by IDT and cloned into the base vector containing the gRNA and protein machinery required to obtain targeted insertion (+ORF1, +ORF2-Cas9) through In-Fusion reactions.

To test the delivery of the herbicide bialaphos resistance gene (*bar*)[57], multiple inserts were amplified for In-Fusion reactions. DNA fragments of the *bar* CDS and *NOS P–bar–NOS T* were amplified from pB2GW7, embedded in *mPing* and fused into the base vector containing the gRNA and protein machinery required to obtain targeted insertion (+ORF1, +ORF2, +Cas9).

For the HITI experiment, two gRNAs (one targets *mPing* upstream of *ACT8*; the other cuts the flanking sequence on either side of *mPing*, excising the entire *mPing* element) were added into the Cas9-only vector through a GoldenGate reaction. *mPing* with flanking gRNA target sequences (targeted by the gRNA mentioned above) was amplified and added to the vector through In-Fusion reactions.

The soybean vectors are based on pTF101.2. The *mPing* donor, gRNA cassette, and ORF1 and ORF2-Cas9 were amplified from the above vectors constructed for *Arabidopsis* transformation and cloned into pTF101.2 through In-Fusion reactions. The *DD20*[58] gRNA sequence was added on the overhangs of In-Fusion primers.

## Plant growth and transformation

Wild-type *Arabidopsis* plants (Columbia ecotype) were grown at 22 °C on Pro-Mix FPX soil in Conviron MTPS-120 growth chambers with 16 h per day of 200 µmol m$^{-2}$ s$^{-1}$ light. Binary vectors were introduced into *Agrobacterium tumefaciens* GV3101. All of the transgenic lines were transformed using the *Agrobacterium*-mediated floral dip method and subsequent selection for hygromycin-resistant plants.

Soybean plants were transformed by the Plant Transformation Facility at the Donald Danforth Plant Science Center. Binary vectors were introduced into *A. tumefaciens* AGL1. Mature half-seeds of Soybean (*G. ma*x var. Williams 82) were transformed using *Agrobacterium* to generate transgenic plants using methodology adapted from a previous study[59]. Transgenic plants were selected for Basta resistance and confirmed by PCR. Soybean plants were grown at 25 °C during

the day and 23 °C during the night in growth chambers with 14 h per day of 200–600 µmol m$^{-2}$ s$^{-1}$ light, and transferred to a greenhouse at 25 °C during the day and 23 °C during the night.

## Determination of *mPing* excision

Images of *Arabidopsis* seedlings were captured using the Axio Zoom.V16 microscope (ZEISS) with a PlanNeoFluar Z ×1.0 objective. The excitation wavelength was 450–490 nm, and the emission wavelength was 500–550 nm. The *mPing* excision was also evaluated by PCR analysis (Fig. 1b), with the primers listed in Supplementary Table 1.

## Determination of Cas9 mutation rate

The T7 endonuclease I-based mutation detection assay (NEB) was used to access the Cas9 mutation rate. The targeted DNA regions were amplified using Q5 High-Fidelity DNA Polymerase (NEB). A list of the PCR primers used in these reactions is provided in Supplementary Table 1. Heteroduplex formation and T7 endonuclease I (NEB) digestion were performed on the amplified PCR products, and digested products were then visualized by agarose gel electrophoresis.

## Determination of targeted insertion

For each gRNA-targeted site, four PCR reactions were conducted as described in Fig. 1c. A list of the PCR primers used in these reactions is provided in Supplementary Table 1. The PCR products of the expected size were then Sanger sequenced to confirm the presence of the targeted insertions. In some cases the PCR products were cloned into the pCR4_TOPO TA vector (Thermo Fisher Scientific) and sequenced using purified plasmids from single colonies. All of the Sanger sequencing data were aligned and visualized using the Benchling Biology Software.

## Amplicon-seq of targeted insertion at *PDS3*

PCR amplicons were generated using the primers shown in Supplementary Table 1. For the amplicon sequencing (amplicon-seq) in Fig. 2, the placement of primers is shown in Fig. 2b. Sequencing libraries were constructed from the amplicons using the Nextera DNA Flex kit (Illumina) and Hackflex protocol[60], as described previously[61]. After quality control, the library with pooled amplicons was sequenced with single-end 300 bp reads on the Illumina MiSeq system at the University of Delaware DNA Sequencing & Genotyping Center. 3′ adapter sequences were removed using cutadapt[62] (parameters: -a CTGTCTCTTATACACATCT -m 10). Bioinformatic analyses are described below.

## Bioinformatic analysis of insertion-site precision

Sequencing runs were managed in and initially processed using Illumina BaseSpace software. To determine the precise site of *mPing* integration within *PDS3* from amplicon-seq data, first the *mPing* sequence was identified and removed using cutadapt[62] (parameters: -a (or -g) "XXX....XXX;min_overlap=8" --discard-untrimmed). The 20 nucleotide *mPing* sequence used as the adapter was different if the orientation of insertion is forward or reverse and whether the left or right junction is investigated. Similarly, the *mPing* sequence was searched for on the 5′ (-g) or 3′ (-a) depending on which junction was investigated. After removing the *mPing* sequence, the rest of the sequence was mapped to the *PDS3* reference sequence with the flanking TTA at the targeted location using bwa mem[63] (default parameters). The left-most or the right-most base pair location of the mapped read (depending on right or left junction investigation) was reported as the insertion site. The flanking TTA was included in the reference sequence to ensure that presence or absence of flanking TTA does not impact the determination of the insertion site.

## Bioinformatic analysis of *mPing* intactness

To determine whether the full-length *mPing* was delivered to the target site, we investigated the reads that contain the *mPing/PDS3* junction defined as having both the ≥20 nucleotide *PDS3* border sequence and

≥20 nucleotide *mPing* sequence that is bordering *PDS3*. Cutadapt[62] parameters to identify these sequences were non stringent to allow for imperfect insertion sites to be included in the analyses. First, cutadapt was used to identify and remove the *PDS3* sequence (parameters: --discard-untrimmed --rc --action=trim -g (or -a) "XXX…XXX;min_overlap=8;e = 0.11"). Second, cutadapt was used to identify the 20 nucleotide *mPing* sequence and convert the full *mPing* sequence to lowercase using parameters: --discard-untrimmed –rc –action=lowercase -a (or -g) "XXX…XXX;miin_overlpa=8;e = 0.11;anywhere"). Lowercasing helped identify the *mPing* sequence away from the flanking TTA sequences that could impact the downstream mapping. The full *mPing* sequence was then extracted and mapped to the corresponding reference sequence using the default parameters of bwa mem[63]. The start (or end) positions of all mapped reads were counted and displayed in Fig. 2b. Forward and reverse insertion orientations of *mPing* were merged.

## Investigation of junction nucleotides at the site of targeted insertion

Reads were mapped to the expected targeted sequence with flanking TTA separating *PDS3* border sequence and the inserted *mPing* sequence using bwa mem[63] with the default parameters. Haplotype-Caller from the Gatk toolkit[64] was then used to identify all of the variants−insertions, deletions and SNPs across the target sequence (parameters: --max-reads-per-alignment-start 0 --disable-tool-default-read-filters -ERC BP_RESOLUTION). For Fig. 2d, only the variants in the junction between *mPing* and *PDS3* are shown. Data for forward and reverse orientation of *mPing* insertion were merged and R package ggplot2 was used for the data display.

To calculate the statistical difference between the level of polymorphism at the target site between the fused +ORF2-Cas9 and unfused +ORF2, +Cas9 configurations, first the percentage of reads reporting 0 polymorphisms at each position was calculated. Then, the distribution of this percentage across the junction was compared between the fused and unfused configurations and Pearson's $\chi 2$ tests were used to test statistical significance. In Fig. 2d, the difference between the two configurations was found to be statistically significant for both left ($P < 0.01$) and right ($P < 0.001$) junctions.

As Tn5 was used to add the Illumina adapters after PCR amplification, it is impossible to detect PCR duplicates and the number of unique insertions therefore cannot be calculated. Thus, we determined the minimum number of insertions by counting the number of reads with unique sequences after clustering the reads, that is, reads that overlap with exact match in the overlapping region would be merged to be called as a single insertion. This method will underestimate the number of unique insertions. First, all of the reads that cover the junction shown in Fig. 2d are collected. These reads are then clumped to remove exact duplicates using bbduk clumpify.sh[65] (parameters: containment=t subs=0 addcount=t dedupe=t). Finally, clusters were called based on overlapping reads also using bbduk dedupe.sh (parameters: storename=f uniquenames=f Sort=length absorbrc=t absorbmatch=t absorbcontainment=t findoverlap=t cluster=t processclusters=t cc=t exact=t minoverlap=20 k = 20). The number of unique clusters was counted for each sample and shown in Fig. 2d.

## Determination of *mPing* free-transposition sites

Libraries for insertion-seq were constructed using an adapted HtStuf protocol[66] with the following modifications to reduce the abundance of the *mPing* donor site in the library. High molecular mass DNA was isolated from 50 *Arabidopsis* seedlings for each line using the NucleoBond HMW DNA kit (Takara), and digested by the restriction enzymes XbaI and AluI (sites not present in *mPing*). The XbaI enzyme was selected because the donor *mPing* position has XbaI sites just outside of *mPing*, generating a fragment of 446 bp. Fragmented DNAs above 450 bp were purified from agarose gels, A-tailed by Klenow fragment (3′−5′ exo-; NEB) and ligated to the pGEM-T Easy vector (Promega). Then, 1 µl of this ligation product was used as a template for primary PCR, followed by secondary PCR using nested primers. *mPing*-specific primers were used with the pGEM-T Easy vector primers for primary and secondary PCRs. Barcoded sequencing adapters were added to the amplicons through their inclusion in the PCR primers. A list of all of the primers used is provided in Supplementary Table 1. PCR products were purified using the DNA Clean & Concentrator kit (ZYMO Research) and pooled. After passing quality controls, libraries were run on the MiSeq (Illumina) system with V2 output (single-end, 300 bp) at the University of Delaware DNA Sequencing & Genotyping Center. The sequencing library for soybean was constructed similarly, except that the genomic DNA was isolated from leaf tissues and digested by XbaI, PmlI and AluI.

## Bioinformatic analysis to determine free-transposition sites

Sequencing runs were managed in and initially processed using Illumina BaseSpace software. To identify and characterize reads that have both *mPing* and other regions of the genome, the 3′ adapter was trimmed from raw reads using cutadapt[62] (parameters: -a "ATCACTAGTGAATTCGCGGCC;min_overlap=10;e = 0.1" -q 10). Next, only reads containing *mPing* sequence were identified and the matching *mPing* sequence was trimmed from the 5′ end using cutadapt (parameters: -g "EXPECTEDMPINGSEQUENCE;min_overlap=35;e = 0.1" -q 10). Reads shorter than 30 nucleotides were discarded. To remove reads that show *mPing* at its donor location, these 5′ and 3′ trimmed sequences were mapped to the reference *mPing* donor sequence using the default parameters of bowtie2[67] with the additional parameter to store donor-unmapped reads (--un). These donor-unmapped reads were then mapped to the genome (TAIR10 reference genome for *Arabidopsis* and Williams 82 reference genome Wm82.a4.v1 from Phytozome for soybean) using the default parameters of bowtie2. The start position of each mapping read was collected as the *mPing* insertion site into the genome. The counts for insertion sites were summed over 10 nucleotide non-overlapping bins across the genome and normalized to the sequenced raw read counts in each sample to calculate the RPM for each ten-nucleotide bin. The R package CMplot[68] was used to generate the Manhattan plots displayed in Figs. 2e and 4d and Extended Data Fig. 6b.

The number of bins with ≥100 RPM in each sample was counted as the number of free-transposition sites in the genome. *PDS3* bins were excluded from the free-transposition list for samples with the *PDS3* gRNA. The overlap of free transposition between biological replicates is shown in Extended Data Fig. 6a. The free-transposition sites were sorted on the basis of their RPM values and shown on the *x* axis in decreasing order in Fig. 2f with the RPM plotted. To only interrogate sites with partial matches to gRNA, first a list of *Arabidopsis* genomic regions was created using Cas-OFFinder[69] with the least stringent criteria (≤9 bp mismatch to the gRNA and ≤2 bp bulge size). Next, only overlapping bins of free-transposition sites were retained and the Manhattan plots were created for Extended Data Fig. 6b.

## Western blotting

In liquid nitrogen, inflorescence tissue was ground into fine powder, then thawed in lysis buffer (50 mM Tris-HCl pH 7.5, 150 mM NaCl, 5 mM MgCl$_2$, 10% glycerol, 1% NP-40 (IGEPAL), 0.5 mM DTT, 1 mM PMSF, 1% plant protease inhibitor cocktail (GoldBio)) for 15 min at 4 °C. The lysate was centrifuged for 15 min at 4 °C to remove debris. Equal amount of 2× LDS sample loading buffer (NuPAGE) was added to the clarified lysate. The samples were incubated at 95 °C for 5 min and then loaded onto a 4−20% gradient Tris-Glycine gel (BioRad). Proteins were separated at 150 V for 75 min and then transferred to a PVDF membrane (Immobilon-FL, MilliporeSigma) using the BioRad semi-dry transblot for 1 h. The membranes were blocked for 1 h at room temperature in Azure Fluorescent Blot Blocking Buffer (Azure). Primary antibodies, anti-Actin 11 (Agrisera, AS10 702) and anti-Cas9 (Diagenode, C15310258-100), were diluted 1:2,000 and 1:5,000, respectively, in the blocking buffer and incubated with the blot for three nights. The membranes

were washed five times at room temperature with 1× PBS-T. The secondary antibodies (anti-Actin 11: AzureSpectra, goat anti-mouse 800, AC2135; anti-Cas9: AzureSpectra goat anti-rabbit 800, AC2134) were diluted 1:5,000 and incubated with the membranes for 1 h. The membranes were washed five times at room temperature with 1× PBS-T, and with 1× PBS for the last wash. The blots were dried and visualized using the Azure Sapphire Biomolecular Imager. Raw images of the western blots are shown in Supplementary Fig. 1.

### HRM analysis to detect mutations

High resolution melt (HRM) analysis was used to detect mutations at specific cut sites by Cas9. A list of the primers used for the analysis is provided in Supplementary Table 1. PCR reactions were performed using the QuantStudio 5 real-time quantitative PCR (qPCR) system and MeltDoctor HRM reagent (Applied Biosystems) according to the manufacturer's instructions. The qPCR data were then analysed using the High Resolution Melt Software from Applied Biosystems.

### DNA methylation analysis

DNA methylation was analysed using DNA isolated from *Arabidopsis* seedlings using the NucleoBond HMW DNA kit (Takara). High-molecular-mass DNA was digested with XbaI and HincII or XbaI, DraI and SmaI to enrich for targeted integrations and remove donor locations of *mPing*. Digested high-molecular-mass DNA was run on an agarose gel and fragments above 1 kb were extracted. Fragmented DNA was converted (unmethylated Cytosines enzymatically converted to uracil) using a modified protocol of NEBNext Enzymatic Methyl-seq Conversion Module (New England Biolabs). In brief, the modifications included: 3 times the suggested volume of enzyme TET2 and oxidation enhancer were used to compensate for an increased input of 500 ng of genomic DNA; and, in the DNA clean-up steps, ethanol precipitation was used rather than bead purification. Amplicons were generated for sequencing by PCR for each locus with primers including degenerate bases that can be found in Supplementary Table 1. Amplification was performed using the My Taq HS Mix (Bioline) and the correct-size band was then extracted from the agarose gel. To calculate the conversion rate of each enzymatically converted DNA sample, amplification was also performed on *AT2G20610*, a known unmethylated gene. Library preparation and amplicon-seq was performed as described in the 'Amplicon-seq of targeted insertion at *PDS3*' section above. Analysis was performed as previously described[61].

### Yeast assays

Yeast transposition assays were performed as described previously using BY4741-derived yeast that contain a genomic copy of the *mmPing20* element in the *ADE2* gene[49]. ORF1 (ORF1SC1 ONE[70]), ORF2 (*Pong* TPase L418A, L420A)[71] and Cas9 were supplied separately or as fusions cloned into pAG423 GAL (Addgene, 14149) and pAG425 GAL (Addgene, 14153) plasmids using standard gateway cloning. p426-SNR52p-gRNA.CAN1.Y-SUP4t (Addgene, 43803) was used to supply the CAN1 gRNA[72]. The number of *ADE2* revertant colonies were counted after 10 days incubation on Galactose CSM-ADE-HIS-LUE-URA. A total of 48 colonies were grown on a CSM-ADE plate, then replicated onto CSM-ARG + canavanine medium and scored for growth the next day.

### Statistics and reproducibility

Statistical analysis for individual experiments is described within their figure legends. Figure 1a and Extended Data Fig. 2b are representative images from 160 independent plants (10 from each of the 16 lines) that were analysed with similar results. The analysis of *mPing* excision (Fig. 1b and Extended Data Fig. 2c) and targeted integration (Fig. 1d and Extended Data Fig. 4a) was repeated with independent transgenic plants twice with the same results. The Western blots in Extended Data Fig. 4b were repeated twice. Excision assays of *mPing* with Cas12a fusion proteins (Extended Data Fig. 5b) and targeted insertion assays (Extended

Data Fig. 5e,f) were independently performed with two pooled samples (each including 50 independent $T_1$ transgenic plants) with similar results. The targeted insertion assay with the one-component transgene in Extended Data Fig. 7c was repeated twice with similar results. The excision assay for *mPing_HSE* (Extended Data Fig. 8b) and the targeted insertion assay for this element (Extended Data Fig. 8c) was repeated with independent transgenic plants twice with similar results. The excision assay for *mPing_bar* and *mPing_bar_CDS* (Extended Data Fig. 9a) and the targeted insertion assay for these elements (Extended Data Fig. 9c) was repeated using pooled samples (each including 30 independent $T_1$ transgenic plants) twice with similar results. The PCR genotyping in Extended Data Fig. 12b was repeated for two technical replicates on the same plants.

### Biological material availability

The LbCas12a plasmid and sequences are from Bayer Crop Science and are not controlled or distributed by the corresponding author. Otherwise, there are no restrictions on the availability of the biological materials. Materials are available from the authors and the Cas9 plasmid vectors and are being made available from the Arabidopsis Biological Resource Center (http://abrc.osu.edu).

### Reporting summary

Further information on research design is available in the Nature Portfolio Reporting Summary linked to this article.

## Data availability

There are no restrictions on the presented data. Amplicon-sequencing and Insertion-seq data from Figs. 2 and 4 for *Arabidopsis* and soybean are provided through the NCBI Sequence Read Archive (GSE227105). Genome sequences and annotations used come from TAIR10 (Columbia ecotype *Arabidopsis*) (https://www.arabidopsis.org/download/) and Williams 82 Wm82.a4.v1 from Phytozome (soybean; https://phytozome-next.jgi.doe.gov/info/Gmax_Wm82_a4_v1).

## Code availability

All tools used for the bioinformatic analyses are publicly available. Unless specified otherwise, the default parameters were used.

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

**Acknowledgements** We thank K. Czymmek for help with microscopy; T. Finley, P. Vallal, M. O. Luis and D. Arasu for supporting the soybean transformation workflow; S. Wessler for her comments; and the staff at the University of Delaware DNA Sequencing & Genotyping Center. This project is supported by NSF grants IOS-2149964 to R.K.S. and MCB-1651666 to C.N.H., the Danforth Plant Science Center Proof of Concept funding, The Danforth Plant Science Center Big Ideas Competition and a sponsored research project from Bayer Crop Science.

**Author contributions** The concept was developed by P.L. and R.K.S. Methodology was developed by P.L., C.N.H. and R.K.S. Bioinformatics analysis was developed by K.P. The investigation was carried out by P.L., K.P., S.A.E., P.P., Y.-H.H., R.S., K.N.R. and G.K. Data curation was the responsibility of P.L., K.P. and R.K.S. Soybean transformation was performed by V.V. and H.Y. Design of Cas12a vectors and analysis was aided by X.Y. and L.A.G. Yeast assays were performed by M.V.C. and C.N.H. Materials were provided by C.N.H.; P.L. and R.K.S. wrote the paper. Funding was acquired by R.K.S.

**Competing interests** Part of the funding that supported this research in the Slotkin laboratory was provided by Bayer Crop Science. X.Y. and L.A.G. work to improve crop gene editing for the for-profit company Bayer Crop Science. Patents have been filed with inventors R.K.S. and P.L. at the Donald Danforth Plant Science Center, and the technology is licensed to Bayer Crop Science for research purposes. The following patents have been filed, with R.K.S. and P.L. listed as inventors on both: (1) "Targeted insertion via transposition"; pending application numbers, US18/282,139, EP22772096.8, CA3,212,093 and AU2022237499A1; applicant, Donald Danforth Plant Science Center. (2) "Targeted insertion via transposition"; pending application number PCT/US2023/078837; applicant, Donald Danforth Plant Science Center.

**Additional information**
**Correspondence and requests for materials** should be addressed to R. Keith Slotkin.

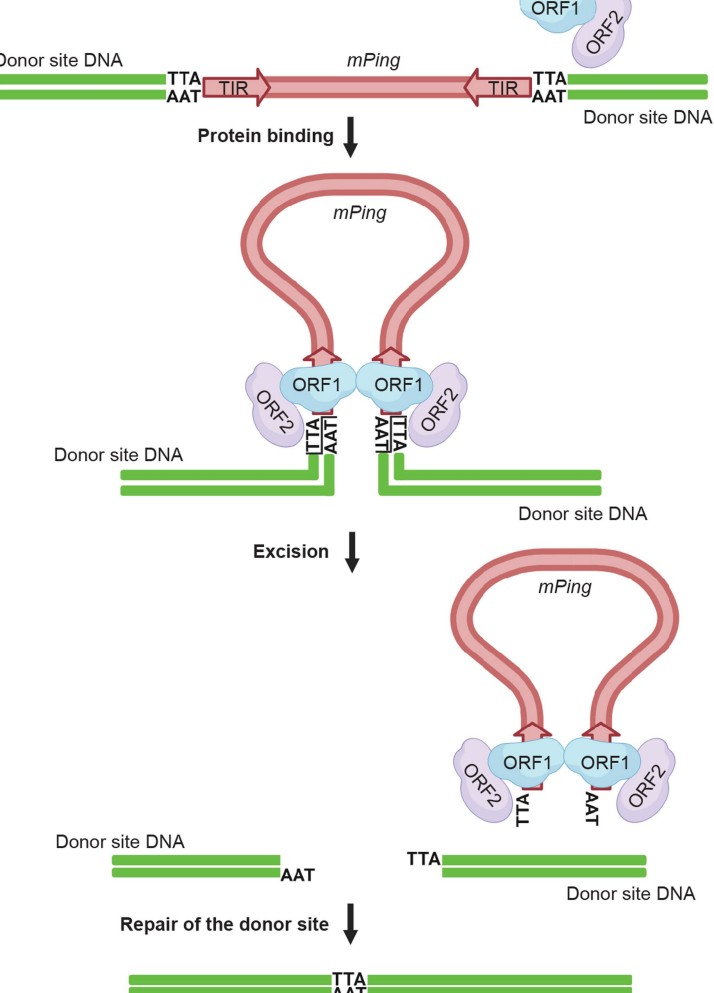

**Extended Data Fig. 1 | Published data in support of the model of *mPing* excision.** The ORF1 and ORF2 proteins are expressed from the *Pong* transposon and bind the *mPing* element to form a transposition complex[31,73]. ORF1 is a Myb-like DNA binding protein that binds to at least 15 base pairs of the *mPing* terminal inverted repeat (TIR) sequence[49]. ORF2 is the canonical transposase (TPase) with the DDE catalytic motif necessary for *mPing* excision and insertion[71,74]. The flanking nucleotides (TTA or TAA) that are immediately adjacent to the TIRs at the donor site are necessary for efficient *mPing* excision[32]. The ORF1 and ORF2 proteins directly interact[75] and are both required for *mPing* excision from the donor site[9,71]. After excision, the donor site is repaired by NHEJ using the microhomology of the staggered cut overhangs left by excision[32]. This allows for very precise repair of the excision site, often reestablishing the coding frame of the *mPing* donor site[71,76,77]. The transposition complex remains associated with the extra-chromosomal *mPing* DNA as it is also responsible for catalysing insertion.

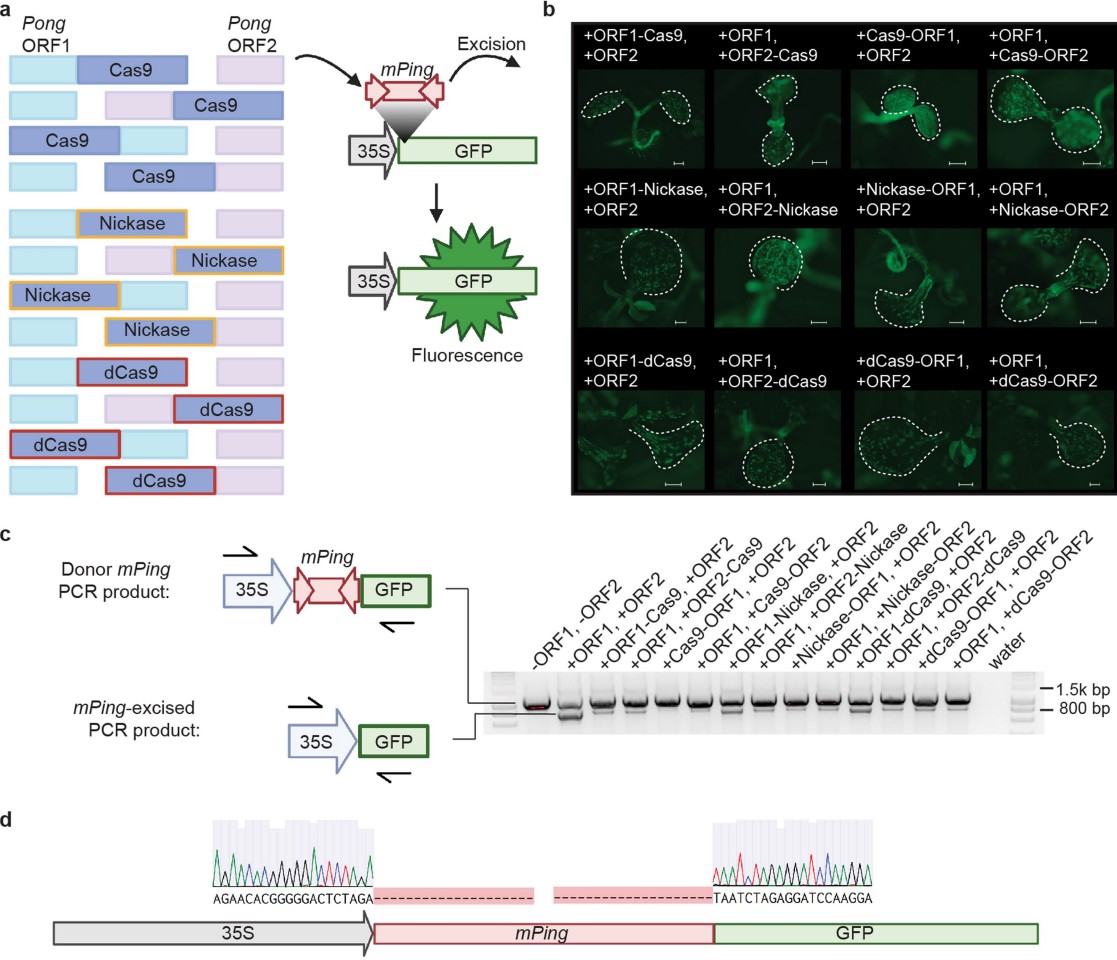

**Extended Data Fig. 2 | Transposable element excision generated by Cas9-fused proteins. a**. Diagram of fusion proteins tested. Twelve different transgenes were created and transformed into Arabidopsis. Cas9 and derivative proteins were fused either to the *Pong* transposase ORF1 or ORF2 protein coding regions. Both N- and C-terminal translational fusions were created using the G4S flexible linker. Three different versions of Cas9 were used: double-strand cleavage Cas9, the single stranded nickase, and the catalytically dead dCas9. When a functional transposase protein is generated by expression of ORF1 and ORF2, it excises *mPing* out of the 35S-GFP donor location in the Arabidopsis genome, producing fluorescence. **b**. Excision of the *mPing* TE from GFP restores the plant's ability to generate fluorescence. Images of representative Arabidopsis seedlings showing GFP fluorescence for all 12 fusion proteins. The cotyledons are outlined with a white dashed line. Size bars represent 500 μm. A subset of this experiment is shown as Fig. 1a. **c**. Excision of *mPing* assayed by PCR of pooled seedlings of the twelve different translational fusions from part **a**, and controls. The top band represents *mPing* within GFP (donor position), and the smaller band is generated upon *mPing* excision. The arrows indicate the pair of primers used for PCR. **d**. Sanger sequencing of the PCR product upon *mPing* excision. Grey bars behind the sequencing peaks represent quality scores for each base call.

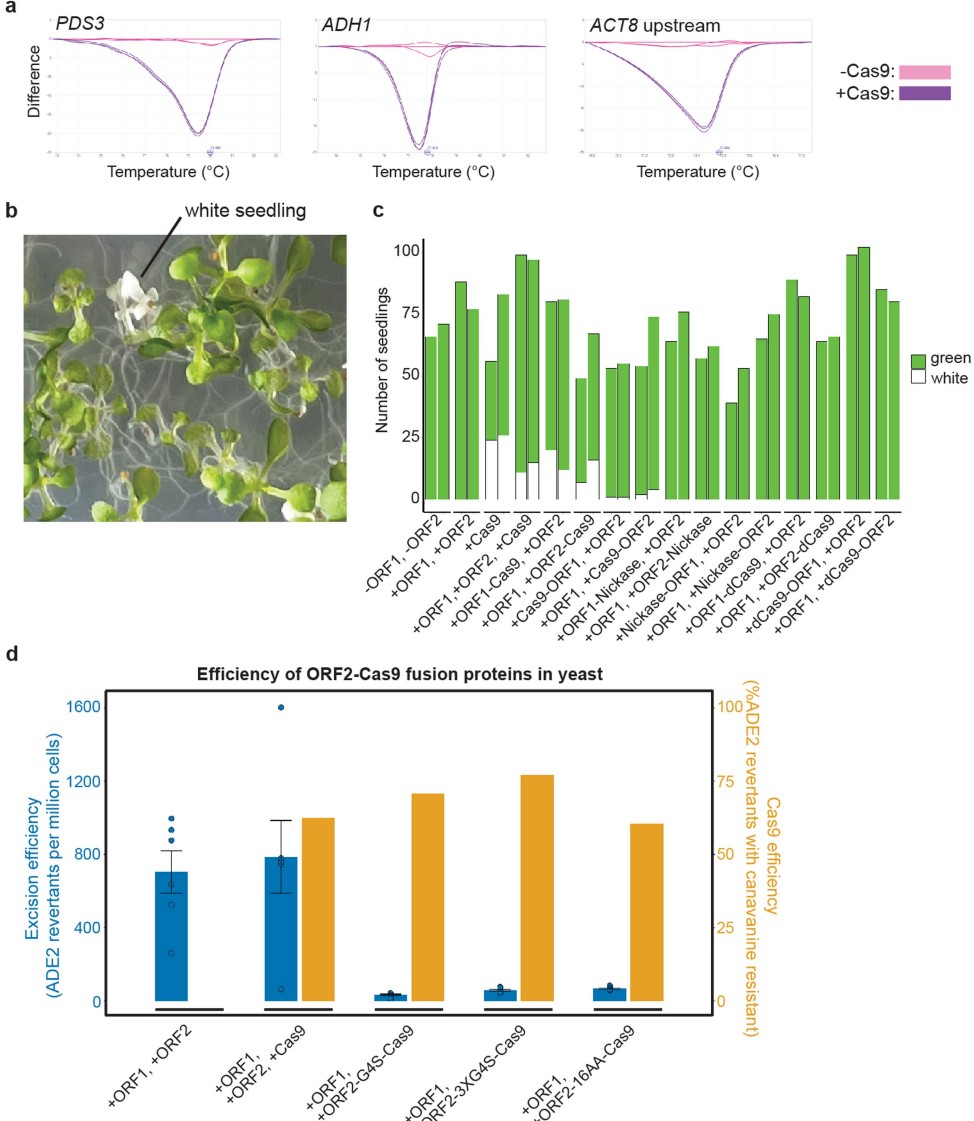

**Extended Data Fig. 3 | Efficiency of ORF2-Cas9 fusion proteins. a.** High Resolution Melt (HRM) analysis to test gRNA efficiency. Mutations created by Cas9 were detected for genomic loci *PDS3*, *ADH1*, or the region upstream of *ACT8*. PCR product melting dynamics differed between the WT plants (pink) and Cas9 positive control lines with the indicated gRNA (purple). The melting temperature difference is caused by the generation of short indels and SNPs upon Cas9 cleavage and repair by NHEJ, verifying that all three gRNAs are functional in Arabidopsis plants. **b.** Representative *pds3* homozygous mutant white seedling from plants with the catalytically-active Cas9 fusion protein. **c.** The ratio of white *pds3* T2 seedlings for all Cas9 fusion proteins tested.

**d.** Efficiency of ORF2-Cas9 fusion proteins in yeast. *mPing* excision frequency (blue, left Y-axis) and Cas9 mutation frequency (orange, right Y-axis) measured for unfused and fused ORF2 and Cas9 with three different protein-protein linker sequences. The ORF2 protein's C-terminus is fused to Cas9 via a 1xG4S, 3xG4S or 16AA linker. *mPing* excision was measured as the number of *ADE2* revertant colonies due to *mPing* excision per million cells. The average and standard deviation for multiple biological replicates (n = 6) are shown (blue). Cas9 mutation frequency was measured by testing the gRNA-targeted canavanine resistance of the *ADE2* revertant colonies (n = 48). This experiment was performed two times independently to ensure reproducibility.

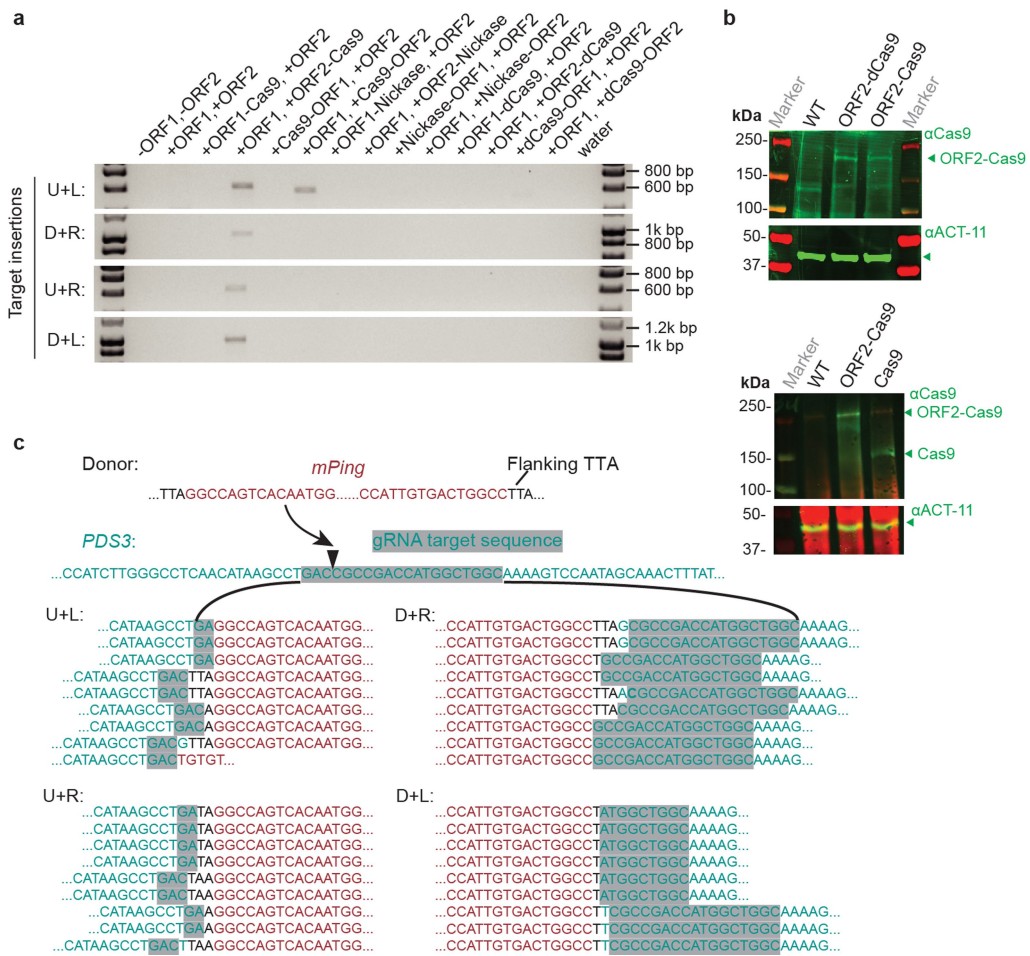

**Extended Data Fig. 4 | Targeted insertions of *mPing* at the *PDS3* gRNA target site. a**. PCR assay as described in Fig. 1c for the 12 fusion proteins generated in Extended Data Fig. 2a and controls. PCR negative controls include a line lacking the Cas9, ORF1 and ORF2 proteins (-ORF1,-ORF2), a line with ORF1 and ORF2 but no Cas9 (+ORF1, +ORF2), and a no-template DNA PCR reaction (water). Among the 12 fusion proteins, only ORF2-Cas9 displays the correct size band for targeted insertions. Insertions were verified by Sanger sequencing of the PCR products. **b**. Western blots using the Cas9 and Actin11 antibodies, showing that the ORF2-Cas9 and ORF2-dCas9 proteins are expressed in transgenic plants as full-length fusion proteins. Upper panel shows that both ORF2-Cas9 and ORF2-dCas9 have the expected size of ~216 kDa (Cas9 is 150 kDa

and ORF2 is 66 kDa). Lower panel compares the size of the unfused Cas9 with the ORF2-Cas9 fusion protein. Raw images of the Westerns are shown in Supplementary Fig. 1. **c**. Sanger sequencing of the junctions of targeted integration events into the *PDS3* gene. PCR products from panel **a** were cloned into the pCR4_TOPO TA vector and 9 individual colonies were sequenced per PCR reaction. The triangle represents where Cas9 cuts in the gRNA target sequence. The flanking TTA/TAA sequence is present at some insertion junctions and absent in others. The *PDS3* sequence is shown in blue, the gRNA target site is highlighted in grey, *mPing* is shown as red text, and the flanking TTA/TAA sequences are shown in black text.

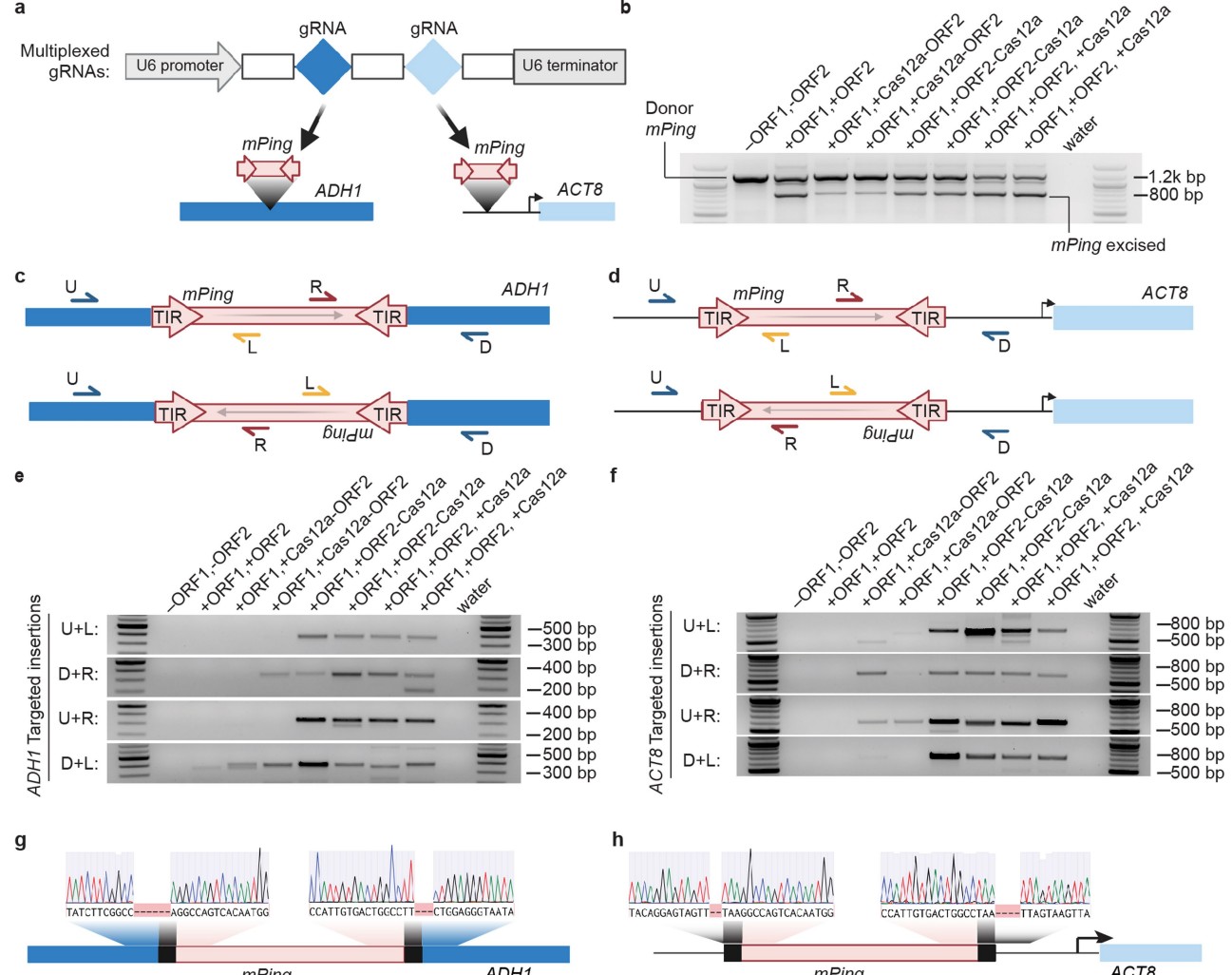

**Extended Data Fig. 5 | Cas12a-mediated targeted insertions. a**. Diagram of the multiplexed vector cassette that generates two distinct Cas12a gRNAs, one that targets *ADH1* and one that targets upstream of *ACT8*. **b**. PCR assay to detect excision of *mPing* generated by functional ORF1 and ORF2 proteins. Fusing these proteins to Cas12a does not stop excision activity. **c**. Diagram of the four PCR reactions to detect targeted insertions into *ADH1*. Arrows indicate primers used to detect targeted insertions: U + L, D + R, U + R, D + L. **d**. Diagram of the four PCR reactions to detect targeted insertions into the region upstream of

*ACT8*. **e**. PCR assay to detect targeted insertion of *mPing* into *ADH1*. Targeted insertions are detected for both protein fusions to Cas12a as well as in the unfused configuration. **f**. PCR assay to detect targeted insertion of *mPing* into the region upstream of *ACT8*. Targeted insertions are detected for both protein fusions to Cas12a as well as in the unfused configuration. **g**. Sanger sequencing of a *mPing* targeted insertion into *ADH1* mediated by Cas12a cleavage. **h**. Sanger sequencing of a *mPing* targeted insertion into the region upstream of *ACT8* mediated by Cas12a cleavage.

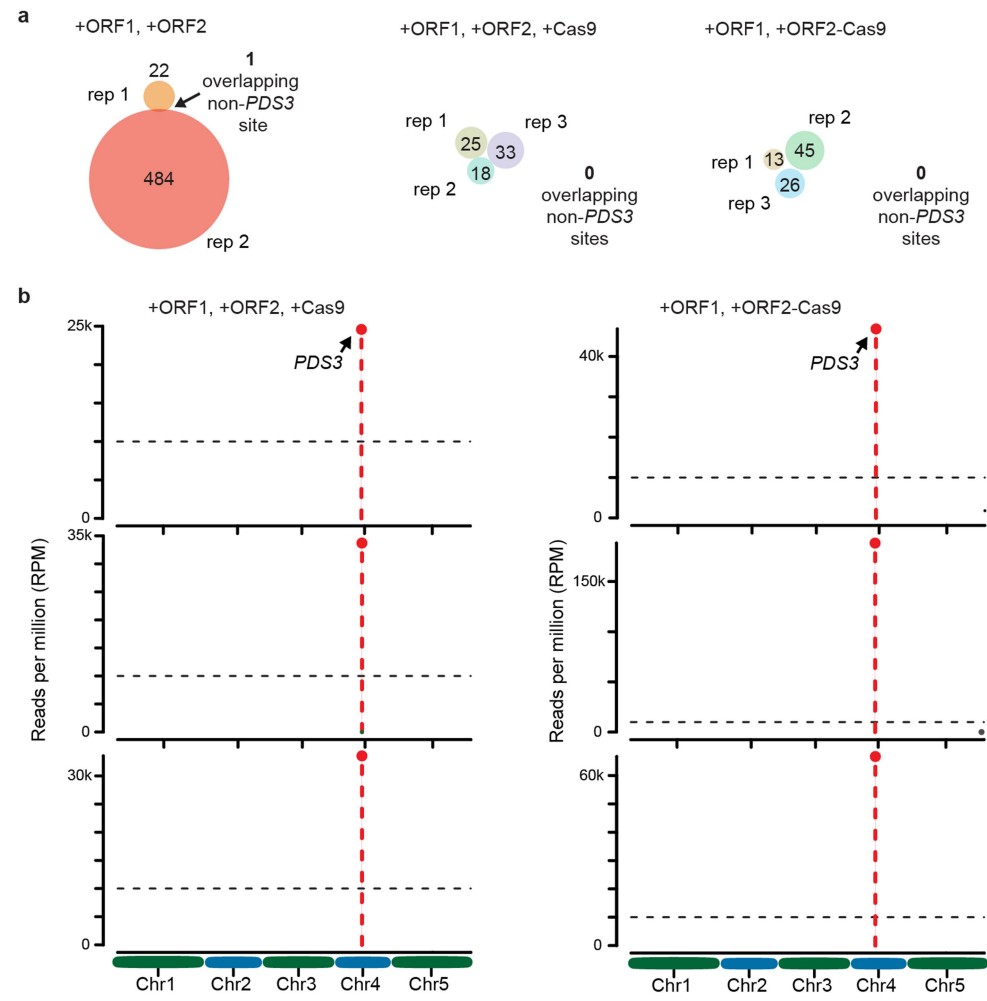

**Extended Data Fig. 6 | *mPing* insertion at gRNA off-target sites. a**. Venn diagrams of *mPing* insertion sites (excluding *PDS3*) in common between biological replicates. Data is from Fig. 2e,f. **b**. Insertion of *mPing* at CRISPR/Cas9 predicted off-target sites. Data display is the same as Fig. 2e. Different from Fig. 2e, only *PDS3* and the predicted CRISPR/Cas9 off-target regions of the genome are interrogated.

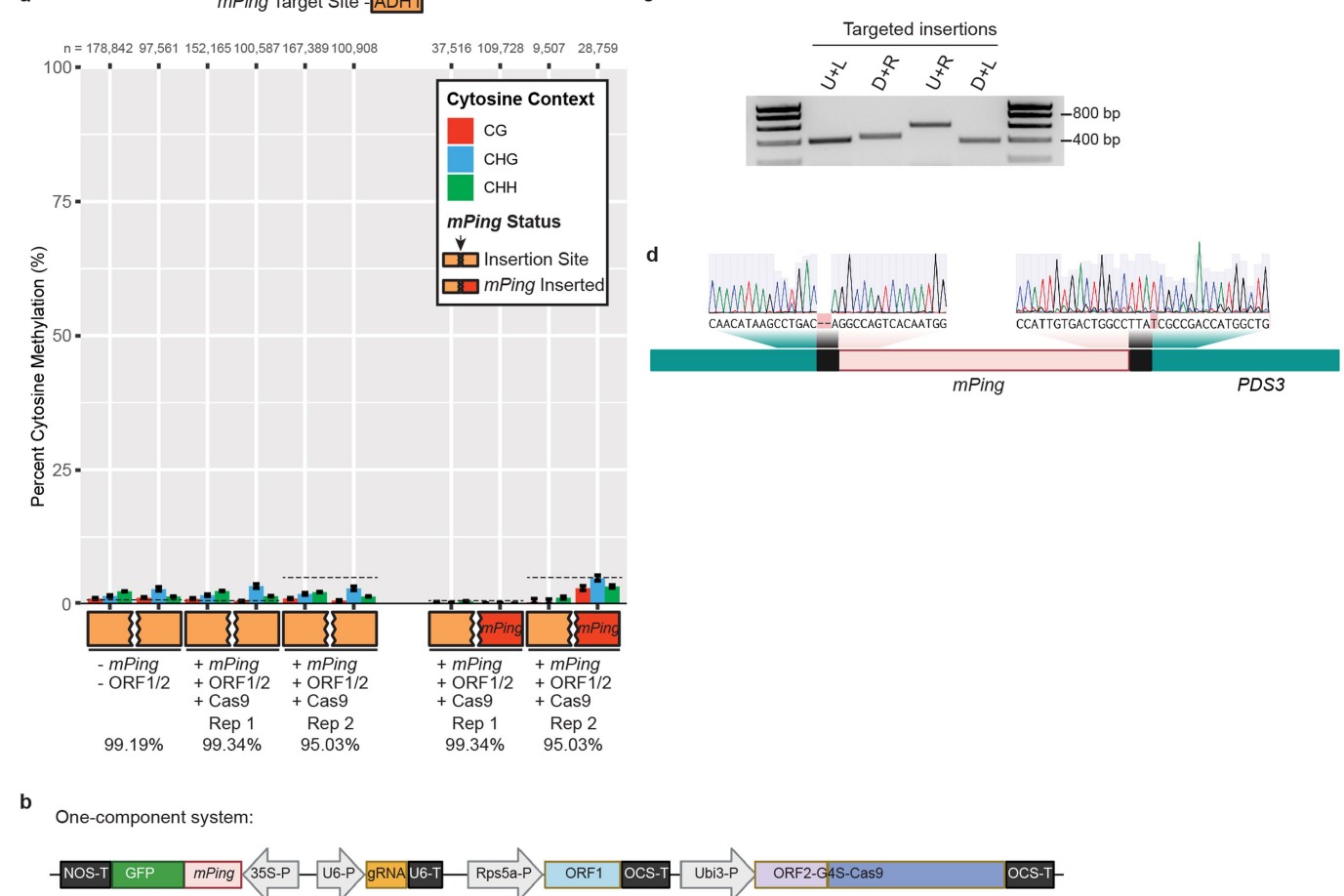

**Extended Data Fig. 7 | DNA methylation analysis and a one-component transgene system. a**. Amplicon deep sequencing of enzymatic-converted DNA methylation patterns. The average methylation level across the amplicon is shown for each cytosine context (CG, CHG, CHH (H = A,T,C)), with 95% confidence intervals calculated using the Wilson score interval method. On the left is the *ADH1* insertion site before *mPing* insertion, broken where *mPing* will insert and either side of the insertion site is analysed separately. On the right is the methylation after *mPing* insertion. A dash line denotes the background non-conversion rate of the enzymatic reaction determined for each sample by sequencing an unmethylated region of the genome. This conversion percentage is also listed below each genotype. Biological replicates are denoted as "Rep 1" vs. "Rep 2". n= the number of total cytosines assayed for each amplicon. **b**. Map of a single vector containing the *mPing* donor element, the gRNA and protein machinery required to obtain *mPing* targeted insertions (+ORF1, + ORF2-Cas9). **c**. PCR-based targeted insertion assay (as in Fig. 1c,d) in pooled seedlings using the one-component transgene system. Targeted insertions are detected in each reaction. **d**. Sanger sequencing of the junctions of a targeted insertion event in the Arabidopsis *PDS3* gene generated from the single vector one-component transgene system shown in panel **b**.

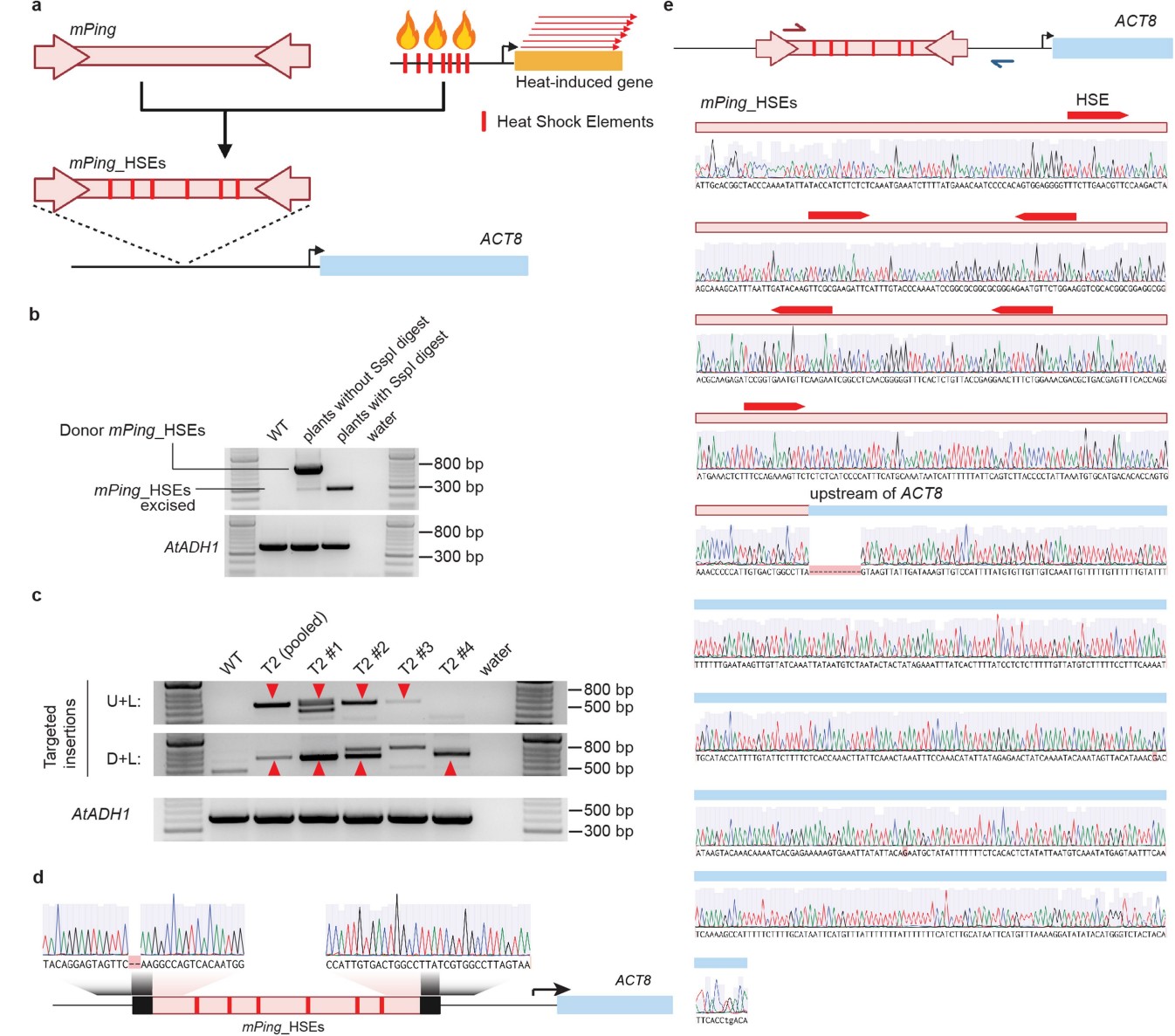

**Extended Data Fig. 8 | Insertion of heat shock elements (HSEs) as *mPing* cargo. a**. Experimental design and generation of the synthetic 444 bp '*mPing_* HSEs' element. **b**. Excision assay by PCR (as in Fig. 1b) in pooled seedlings shows the *mPing*_HSEs element is capable of excision. The excised product is easier to detect if the genomic DNA is digested with the *Ssp*I restriction enzyme before PCR (*Ssp*I site is in *mPing*_HSEs). **c**. PCR assay detecting targeted insertions (as in Extended Data Fig. 5d) of *mPing_HSEs* into the region upstream of the *ACT8* gene. The 'T2 (pooled)' sample represents a pool of T2 seedlings, while 'T2 #1', 'T2 #2', etc... are individual T2 plants. Red arrowheads denote PCR products

that were verified as targeted insertions by Sanger sequencing. **d**. Sanger sequencing of the junctions of a *mPing*_HSEs targeted insertion into the region upstream of *ACT8*. **e**. Sanger sequencing across the majority of the *mPing*_HSEs element and into the region upstream of *ACT8* demonstrates that all six HSEs were successfully delivered to this region. The arrows on the top cartoon indicate the pair of primers used for PCR. The Sanger sequencing represents the contig of several sequencing reactions from a single TOPO TA plasmid clone of a PCR product. The sequence is annotated above, including the six HSEs as red pointed boxes.

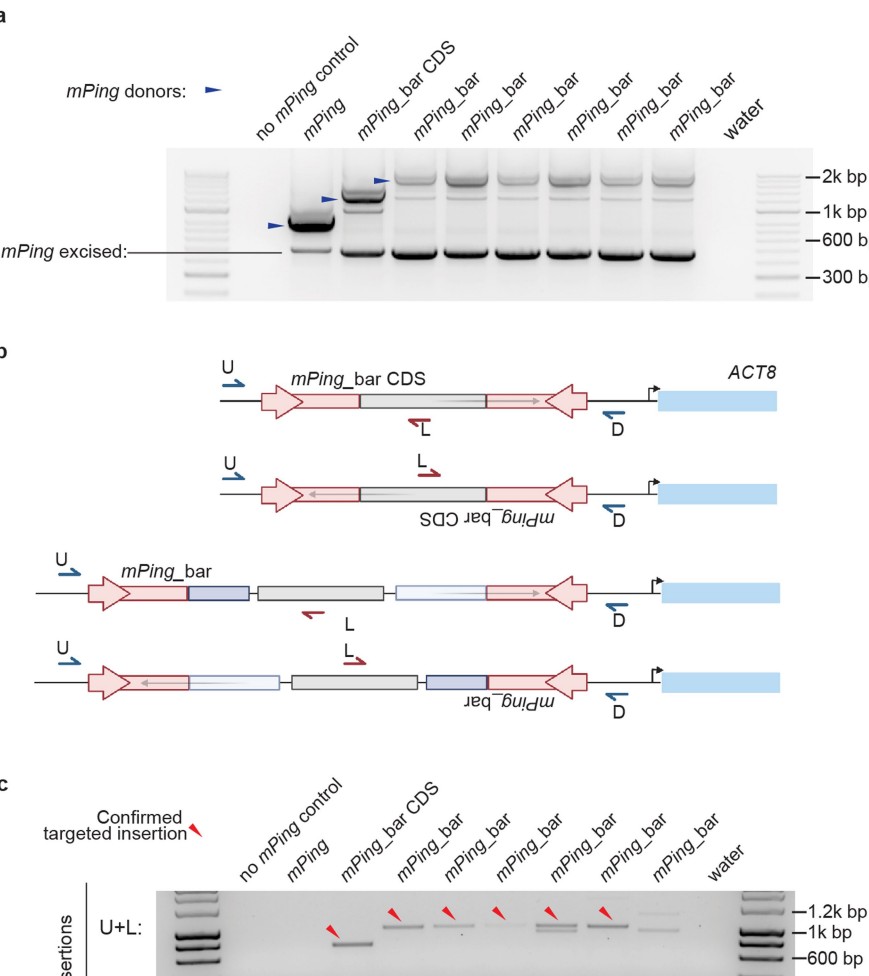

**Extended Data Fig. 9 | Targeted insertion of a gene and CDS as *mPing* cargos. a**. Excision assay by PCR (as in Fig. 1b) in pooled seedlings shows the *mPing*_bar CDS and *mPing*_bar versions are capable of excision. Blue arrowheads indicate the expected size of the amplicon with different sized *mPing* versions before excision. **b**. PCR strategy and primer placement to detect targeted insertions of *mPing*_bar CDS and *mPing*_bar into the region upstream of *ACT8*. Arrows indicate primers used to detect targeted insertions: U + L, D + L. The "L"

primer is the same for *mPing*_bar CDS and *mPing*_bar versions. **c**. PCR detecting targeted insertions of *mPing*_bar CDS *and mPing*_bar into the region upstream of the *ACT8* gene. Red arrows indicate correct size PCR products that were verified as targeted insertions by Sanger sequencing. There is no PCR product in the '*mPing*' sample because the "L" PCR primer site is in the bar CDS region (see panel **b**).

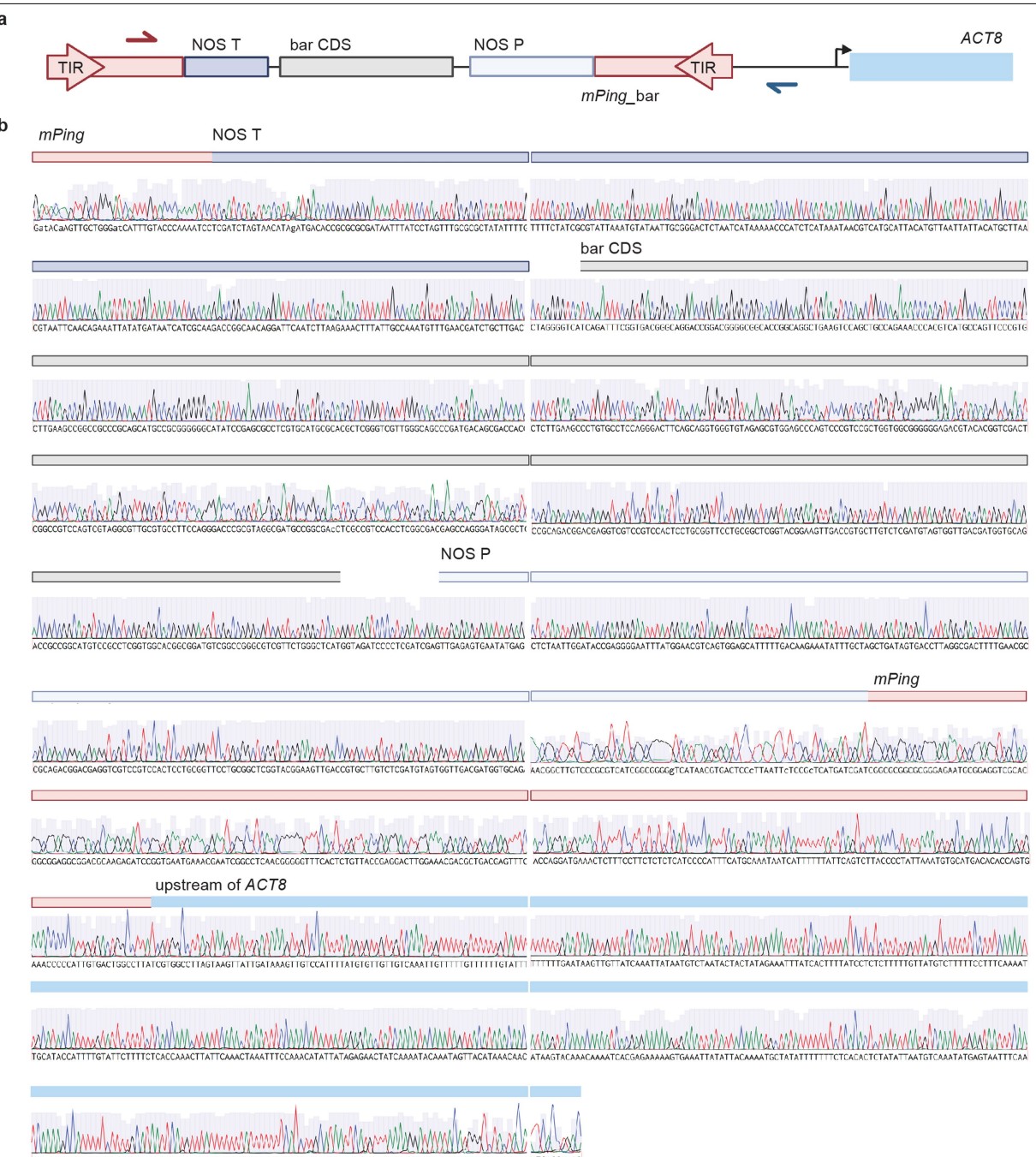

**Extended Data Fig. 10 | Targeted insertion of the intact bar gene cassette.**
a. PCR strategy and primer placement to detect targeted insertions of the *mPing*_bar element into the region upstream of the *ACT8* gene. The arrows indicate the pair of primers used for PCR. b. Sanger sequencing across the majority of the *mPing*_bar element and into the region upstream of *ACT8* demonstrates the successful delivery of the complete bar gene cassette (including promoter and terminator) into this region. The Sanger sequencing represents the contig of several sequencing reactions from a single TOPO TA plasmid clone of a PCR product.

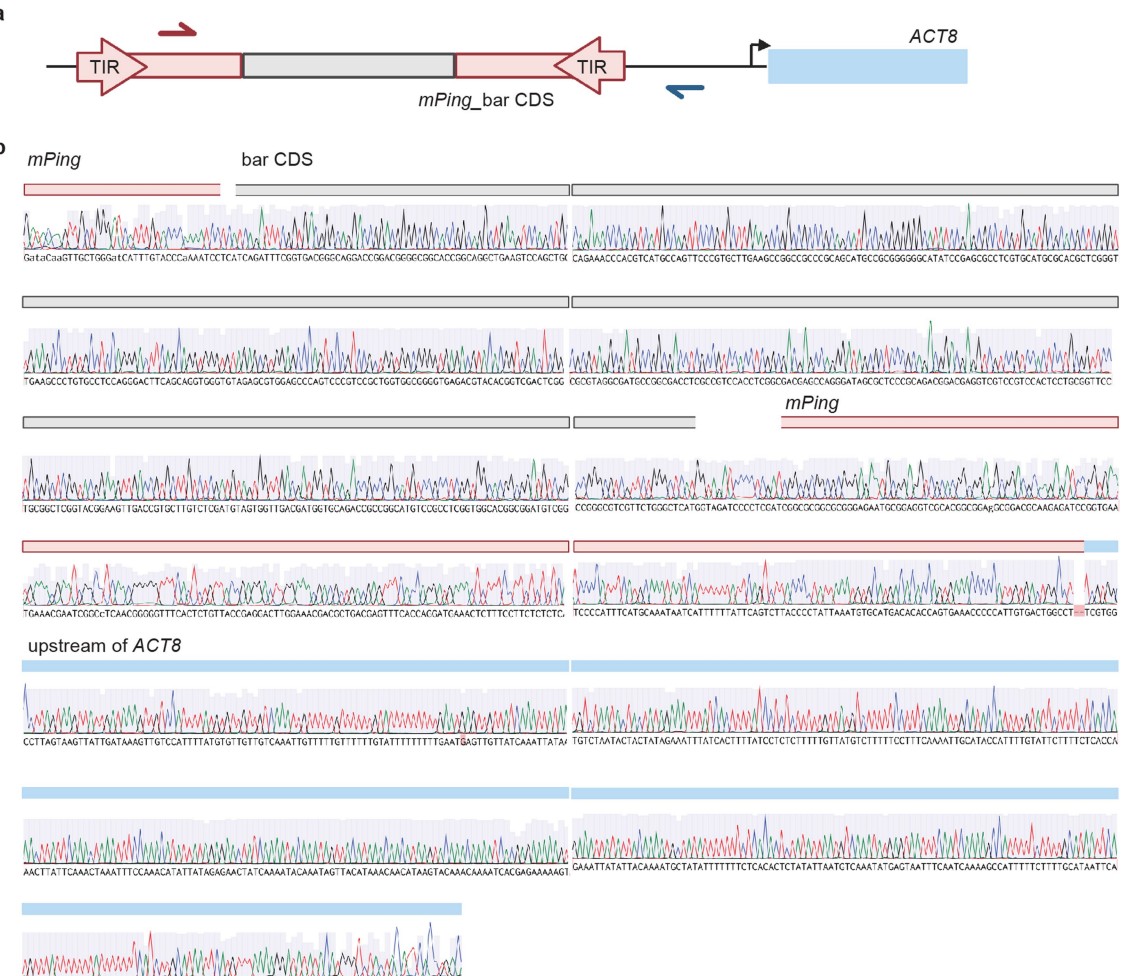

**Extended Data Fig. 11 | Targeted insertion of the intact bar CDS. a**. PCR strategy and primer placement to detect targeted insertions of the *mPing*_bar CDS element into the region upstream of the *ACT8* gene. The arrows indicate the pair of primers used for PCR. **b**. Sanger sequencing across the majority of the *mPing*_bar CDS element and into the region upstream of *ACT8* demonstrates the successful delivery of the complete bar CDS into this region. The Sanger sequencing represents the contig of several sequencing reactions from a single TOPO TA plasmid clone of a PCR product.

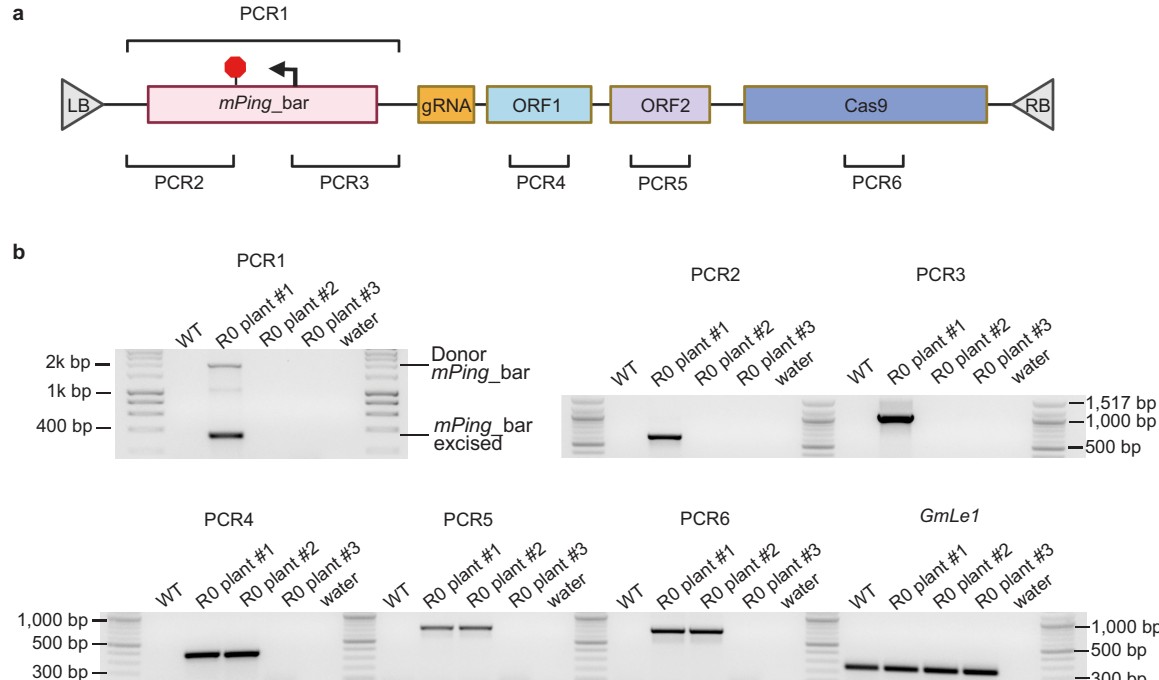

**Extended Data Fig. 12 | The *mPing*_bar element confers the herbicide resistance trait in soybean. a**. Transgene design and PCR primer placement for "PCR1" to "PCR6" used to genotype for the presence of the *mPing*_bar/gRNA/ORF1/ORF2/Cas9 parent transgene in R0 transformed soybean plants. **b**. PCR assay to genotype for the presence of the *mPing*_bar/gRNA/ORF1/ORF2/Cas9 parent transgene in R0 transformed soybean plants. "PCR1" detects both the original *mPing*_bar donor and its excision product. "PCR2" to "PCR6" detect different locations on the *mPing*_bar/gRNA/ORF1/ORF2/Cas9 transgene. *GmLe1* is a control gene. The combined data demonstrates that R0 plant #1 has the full transgene in the genome, plant #2 has a partial transgene insertion that lacks the *mPing*_bar donor site, and plant #3 does not have the *mPing*_bar/gRNA/ORF1/ORF2/Cas9 transgene.

# Reporting Summary

## Statistics

For all statistical analyses, confirm that the following items are present in the figure legend, table legend, main text, or Methods section.

| n/a | Confirmed | |
|---|---|---|
| ☐ | ☒ | The exact sample size (*n*) for each experimental group/condition, given as a discrete number and unit of measurement |
| ☐ | ☒ | A statement on whether measurements were taken from distinct samples or whether the same sample was measured repeatedly |
| ☒ | ☐ | The statistical test(s) used AND whether they are one- or two-sided *Only common tests should be described solely by name; describe more complex techniques in the Methods section.* |
| ☒ | ☐ | A description of all covariates tested |
| ☐ | ☒ | A description of any assumptions or corrections, such as tests of normality and adjustment for multiple comparisons |
| ☐ | ☒ | A full description of the statistical parameters including central tendency (e.g. means) or other basic estimates (e.g. regression coefficient) AND variation (e.g. standard deviation) or associated estimates of uncertainty (e.g. confidence intervals) |
| ☒ | ☐ | For null hypothesis testing, the test statistic (e.g. *F*, *t*, *r*) with confidence intervals, effect sizes, degrees of freedom and *P* value noted *Give P values as exact values whenever suitable.* |
| ☒ | ☐ | For Bayesian analysis, information on the choice of priors and Markov chain Monte Carlo settings |
| ☒ | ☐ | For hierarchical and complex designs, identification of the appropriate level for tests and full reporting of outcomes |
| ☒ | ☐ | Estimates of effect sizes (e.g. Cohen's *d*, Pearson's *r*), indicating how they were calculated |

*Our web collection on statistics for biologists contains articles on many of the points above.*

## Software and code

Policy information about availability of computer code

| Data collection | Illumina BaseSpace software v7.30 |
|---|---|
| Data analysis | High Resolution Melt Software v2.0<br>cutadapt v4.1<br>bwa v0.7.17-r1194-dirty<br>gatk v3.5-0-g36282e4<br>ggplot2 v3.3.6<br>bbmap v39.01 (this covers the bbduk clumpify.sh and dedupe.sh)<br>bowtie2 v2.4.4<br>CMplot v4.2.0<br>Benchling Biology Software v1<br>The perimeters, variables and sequences used for each program are described in the Methods section. |

For manuscripts utilizing custom algorithms or software that are central to the research but not yet described in published literature, software must be made available to editors and reviewers. We strongly encourage code deposition in a community repository (e.g. GitHub). See the Nature Portfolio guidelines for submitting code & software for further information.

## Data

Policy information about availability of data

 All manuscripts must include a data availability statement. This statement should provide the following information, where applicable:

 - Accession codes, unique identifiers, or web links for publicly available datasets
 - A description of any restrictions on data availability
 - For clinical datasets or third party data, please ensure that the statement adheres to our policy

There are no restrictions on the presented data. Amplicon-sequencing and Insertion-seq data from Fig. 2 and 4 for Arabidopsis and soybean are provided via the NCBI sequence read archive as GSE227105. Genome sequences and annotations used come from TAIR10 (Columbia ecotype Arabidopsis)(https://www.arabidopsis.org/download/) and Williams 82 Wm82.a4.v1 from Phytozome (soybean)(https://phytozome-next.jgi.doe.gov/info/Gmax_Wm82_a4_v1).

## Human research participants

Policy information about studies involving human research participants and Sex and Gender in Research.

| | |
|---|---|
| Reporting on sex and gender | N/A |
| Population characteristics | N/A |
| Recruitment | N/A |
| Ethics oversight | N/A |

Note that full information on the approval of the study protocol must also be provided in the manuscript.

# Field-specific reporting

Please select the one below that is the best fit for your research. If you are not sure, read the appropriate sections before making your selection.

☒ Life sciences          ☐ Behavioural & social sciences          ☐ Ecological, evolutionary & environmental sciences

For a reference copy of the document with all sections, see nature.com/documents/nr-reporting-summary-flat.pdf

# Life sciences study design

All studies must disclose on these points even when the disclosure is negative.

| | |
|---|---|
| Sample size | For pools of seedlings analyzed, pools are between 30 and 50 individuals. For the targeted insertion experiment, the number of plants tested is indicated on Figure 3C and 4B. The n= is indicated above the data. No size calculation was performed but rather the sample size was dictated by the number of transgenic plants generated by the experiment. These numbers are sufficient to minimize the effects of ranomd transgene insertion position. |
| Data exclusions | None |
| Replication | Biological replicates (non-overlapping) were used for the deep sequencing data in Fig. 2E-F and Extended Data Fig. 6A-B. |
| Randomization | Samples were allocated into groups based on their genotype and transgene. Within these genotype groups, samples were given random available sequencing indexes. Otherwise, randomization was not relevant to our study. |
| Blinding | Blinding was not relevant to this study because no clinical research was performed. |

# Reporting for specific materials, systems and methods

We require information from authors about some types of materials, experimental systems and methods used in many studies. Here, indicate whether each material, system or method listed is relevant to your study. If you are not sure if a list item applies to your research, read the appropriate section before selecting a response.

## Materials & experimental systems

| n/a | Involved in the study |
|---|---|
| ☐ | ☒ Antibodies |
| ☐ | ☒ Eukaryotic cell lines |
| ☒ | ☐ Palaeontology and archaeology |
| ☒ | ☐ Animals and other organisms |
| ☒ | ☐ Clinical data |
| ☒ | ☐ Dual use research of concern |

## Methods

| n/a | Involved in the study |
|---|---|
| ☒ | ☐ ChIP-seq |
| ☒ | ☐ Flow cytometry |
| ☒ | ☐ MRI-based neuroimaging |

## Antibodies

| | |
|---|---|
| Antibodies used | Primary antibodies: anti-Actin (Agrisera, AS10 702) and anti-Cas9 (Diagenode, C15310258-100). Secondary antibodies: anti-Act 11: AzureSpectra, goat anti-mouse 800, AC2135; and anti-Cas9: AzureSpectra goat anti-rabbit 800, AC2134 |
| Validation | Anti-Actin (Agrisera, AS10 702) is standard in the field as a loading control for Arabidopsis protein on Western blots and is validated at https://www.citeab.com/antibodies/662899-as10-702-anti-actin-11. Anti-Cas9 (Diagenode, C15310258-100) is validated for Cas9 in Arabidopsis in Extended Data Fig. 4b. Cas9 signal is not detected in the WT Arabidopsis plants that do not have a Cas9 transgene. |

## Eukaryotic cell lines

Policy information about cell lines and Sex and Gender in Research

| | |
|---|---|
| Cell line source(s) | BY4741 yeast strain |
| Authentication | *Describe the authentication procedures for each cell line used OR declare that none of the cell lines used were authenticated.* |
| Mycoplasma contamination | *Confirm that all cell lines tested negative for mycoplasma contamination OR describe the results of the testing for mycoplasma contamination OR declare that the cell lines were not tested for mycoplasma contamination.* |
| Commonly misidentified lines (See ICLAC register) | *Name any commonly misidentified cell lines used in the study and provide a rationale for their use.* |

