## [Peer Review file · Nature]

Manuscript Title: Transposase-assisted target site integration for efficient plant genome engineering

Reviewer Comments & Author Rebuttals

Reviewer Reports on the Initial Version:

Referees' comments:

Referee #1 (Remarks to the Author):

In this manuscript, Liu et al. generated a synthetic CAST system by co-expressing catalytically active programmable nucleases with the active rice Pong DNA transposon system and accomplished transposase-mediated targeted insertion in the plants of Arabidopsis as well as of soybean. It is a very interesting and potentially important methodologies, which may help to accomplish targeted integration and delivery of enhancers and gene cargos.

The results are straightforward described and illuminated, however, some data need to be further confirmed or revised.

1. The picture used for Figure 1A-B and Supplemental Figure 1B-D are duplicated, which need to be clarified.
2. The authors compared the fused +ORF2-Cas9 to the unfused +ORF2, +Cas9 configuration, and claimed that their overall precision at the target site was indistinguishable. Whereas, based on the data shown in figure 2D, there are some pattern differences. If a conclusion of 'indistinguishable' to be drawn, statistical analysis need to be performed.
3. The authors claimed that 'The off-target sites in the synthetic CAST lines are not directed by CRISPR/Cas9 cleavage' and 'these off-target sites are a product of the active mPing transposition system and not CRISPR/Cas9 cleavage' because of 'a very few off-target sites are in common between replicates'. The current data did not fully support the conclusion. These part need different controls to make strict comparisons.
4. The effect of the insertion lines either in Arabidopsis and soybean need to illuminated, such as the gene structure and the gene expression changes, and the final phenotype.
5. A case on gene functional modification to show the advantage this technology than others will clearly highlight the importance of this methodology.

Referee #2 (Remarks to the Author):

In their article, Liu et al. exploit the rice transposase, Pong [italics are omitted throughout], alongside the non-autonomous TE mPing, together with a programmable nuclease, Cas9 or Cas12, to achieve targeted integration of DNA payloads ~400-1,500-bp in size at a specific location in the Arabidopsis genome. After demonstrating that Pong, with or without a Cas9 fusion, is still competent for DNA excision of a genomically integrated mPing, the authors design a series of gRNAs to direct site-specific integration in Arabidopsis, as demonstrated by both qualitative PCR, quantitative plant genotyping, and high-throughput sequencing of genome-wide integration specificity. In a last set of experiments, the same strategy is used to achieve targeted insertions in the soybean genome, albeit at much less efficiency.

Altogether, the study is well-written, clearly presented, and the experiments expertly performed; indeed, this work was a pleasure to read. Although this reviewer is not deeply familiar with plant genetics and knock-in methods, the strategy developed by the authors to co-deliver Cas9 and Pong appears to generate much higher efficiencies of targeted knock-ins than have been achieved with other methods in plants, though with the requirement that a mPing payload is first stably (and randomly) integrated into the genome, alongside the genes encoding Cas9 and Pong. A more plant-specific reviewer can better comment on the utility of this approach, as compared to other more standard (non-targeted) integration methods like use of *Agrobacterium*.

From the perspective of genome engineering technology, however, the approach developed by the authors is not distinguished from many earlier strategies that already exploited Cas9 to generate DNA double-strand breaks (DSBs) that increase the rates of homology-independent targeted integration (HITI) of large DNA payloads, in a mechanism that relies on non-homologous end joining (e.g. Lackner et al., *Nat Comm* 2015; Schmid-Burgk et al., *Nat Comm* 2016; Suzuki et al., *Nature* 2016; Serebrenik et al., *Genome Res* 2019). The authors' reliance and requirement for an active nuclease clearly establishes that their approach is similarly dependent on NHEJ-based capture of DNA, and thus the published strategy also overlaps with approaches that have been published and deployed in mammalian cells using alternative transposase proteins such as piggyBac (Luo et al., *Nucleic Acids Res* 2017; Pallares-Masmitja et al., *Nat Comm* 2021; Bazaz et al., *Sci Rep* 2022;) or Sleeping Beauty (Kovac et al., *eLife* 2020). Perhaps most importantly, the authors' own data clearly and convincingly demonstrated that targeted integration works best when Cas9 is not even fused to Pong, indicating that there is almost certain to be no direct synergy or molecular interactions between the RNA-guided targeting component and the transposase-transposon components during the editing step. This conclusion, along with more general semantic considerations (see below), renders the use of cst(CRISPR-associated transposase) inappropriate to describe the present technology.

Admittedly, this work is likely to be the most rigorous of all studies that have used Cas9 in conjunction with a transposase, and is the first (that I am aware of) to do so in plants. Perhaps the most interesting mechanistic/technological value arises from the claim that their high integration efficiencies are enabled because of Pong's ability to protect the DNA ends of the excised payload prior to NHEJ capture, however this claim is not actually proven or supported with experiments. To do so, the authors could compare efficiencies of integration using Cas9 with Pong, versus using Cas9

with additional gRNAs directed to excise the mPing (or any other) payload through additional DSBs, as was developed with HITI and related methods.

Altogether, given that the approach is not new, and that the mechanism of donor DNA integration is not differentiated from earlier studies, I am not sure that this will be of sufficient interest for a general audience to warrant publication in Nature.

MAJOR COMMENTS:

1) The “large, targeted DNA integration” field has gained a lot of attention in recent years, with approaches like HITI, CAST, PASTE, etc., but is at the same time suffering from a lack of detail in the way that technologies are being described and named. The results presented by the authors clearly demonstrates that Pong is functioning as a transposase to excise the mPing genetic payload -- though inactivating mutations to the Pong active site, and/or mutations to delete one or more TIRs from the mPing transposon, would be welcome additions to further prove this point. However, it is clear that DNA integration is NOT occurring through a transposase-based mechanism, but is instead resulting from NHEJ-based capture of the linearized and excised DNA. This can be concluded from the prior literature, from the authors’ experimental results using dCas9, and from the presence of hallmark indel mutations in the integration products.

Thus, it is inappropriate to refer to this technology as a “synthetic CAST,” and all references to this should be removed from the paper. Mis-use of this term will confuse readers, give the impression that a transposase is generating the DNA integration product, and muddy the field.

Separately — even if Pong were fused to dCas9 and being repurposed for the DNA integration reaction — it would still be inappropriate to repurpose the term CAST. CRISPR-associated transposases (CASTs) refer to specific Tn7-like transposons that have naturally coopted nuclease-deficient CRISPR-Cas systems for RNA-guided transposition. If everything involving a CRISPR protein and a transposase becomes a “CAST,” then the meaning and significance of this well-defined term will be sufficiently eroded so as to essentially render it unusable for the field where CASTs are currently and appropriately used.

2) Relating to #1, the use and labeling of TSD (target-site duplication) in Figure 1D, and throughout the rest of the figures, is misleading and should be removed. TSDs are hallmark features that occur upon transposon integration, and they refer to duplications that arise from gap fill-in upon resolution of the transposition product. Here, the authors are mis-using the term to refer to a portion of the flanking sequence that derives *from the original donor site* (where it was a TSD for that first insertion). A TSD is clearly not being generated at the new targeted insertion site, because (a) as established by the authors, these are not even transposition products; and (b) the TSD is rarely duplicated anyways, since their data in Figure 2 show that it is frequently mutated during the NHEJ-based capture step

Although I understand that transposon-flanking sequences are excised alongside the mPing element, as part of the excision reaction — similar to other elements like piggyBac — the authors should modify their terminology to avoid confusing readers into thinking that these are transposition

products. I suggest labeling this as “flanking TTA” or simply “TTA,” and then clarifying better in the text what the origin of this sequence is. (For a general, non-specialist journal, the authors would help out readers by also including a supplementary figure outlining the known chemical steps and sequence features of canonical Pong/mPing transposition, including the way that mPing transposable elements are excised at the TIR boundaries, together with flanking TTA sequences, presumably as 5’ overhangs.)

3) As mentioned above, it would be an interesting and exciting finding, to prove that protection of the donor DNA ends from cellular degradation is a source for the higher integration efficiency in this study, relative to earlier approaches. However, there is no explicit experimental support for statements suggesting that full-length insertions result from “binding of the TE by the transposase proteins while extra-chromosomal, protecting the DNA ends from nucleases and deletions” (line 134-136). I suggest an experiment in which Pong and Cas9 are expressed separately from an mPing derivative that is flanked by two Cas9 cut sites, such that Cas9 expression with three gRNAs will both cleave the target site and excise the mPing payload, similar to HITI. Then, in experiments tested with or without catalytically inactive Pong, the role of Pong in stimulating integration efficiency by protecting the ends – when mPing excision itself is due to Cas9 cleavage activity – can be directly tested and quantified. Importantly, these results would be quite informative for targeted DNA integration applications in other cellular contexts, including for HITI-like applications in mammalian cells, if demonstrated that DNA protection by transposase (or other proteins) increases both DNA integration efficiency and product purity.

MINOR COMMENTS

1) Please add ladder sizes and expected sizes to all electrophoresis gels and Western blot gels throughout the paper.

2) Chromatograms of Sanger sequencing results should be shadowed differently (e.g. in Fig. 1D), so that readers can visualize which regions of the insets correspond to the endogenous PDS3 sequence, the mPing TIR, or the flanking sequence derived from the donor site (i.e. the TTA).

3) The introduction section would be improved by including more general details about the Pong transposase and general Pong/mPing transposition mechanism.

4) The introduction is missing a number of key citations to earlier studies that developed similar approaches to exploit NHEJ-based capture of linear DNA at sites of Cas9-based DNA cleavage. (E.g. Lackner et al., Nat Comm 2015; Schmid-Burgk et al., Nat Comm 2016; Suzuki et al., Nature 2016; Serebrenik et al., Genome Res 2019)

5) Lines 56-60: There are many Class II DNA transposons that do not mobilize through a cut-and-paste mechanism; please update the text accordingly. More importantly, the statement that “TE-encoded transposase proteins... cleave the genome at the integration site” is inaccurate. The DNA is not cleaved as in a hydrolysis mechanism, but is instead subjected to a transesterification reaction, or strand-transfer reaction. This is distinct from what the word “cleavage” implies. (Similar

comment, relating to the use of the word “cut” in line 66.)

6) Figure 1B,C: These panels are difficult to follow without a schematic of the assay provided directly in the same figure. I suggest moving the corresponding panels from the supplement into the main text figure, so that a reader view all the information together.

7) I find the schematic in Fig. 1E to be misleading, related to major comments #1 and #2 above. When viewed in isolation, and/or together with the name “synthetic CAST,” this gives the impression that the Pong transposase proteins are performing the targeted insertion reaction, rather than two completely separate molecular events: DNA DSB formation by Cas9, and NHEJ-based capture of the linear DNA excised by Pong. Fig. 1E and 2C should be combined into a single figure that clearly delineates the likeliest biochemical pathway.

8) The figure schematics and Results/Discussion nearly completely overlook the fact that insertions will be bidirectional using the approach developed by the authors. This will be particularly problematic for experiments in which a gene-of-interest is e.g. integrated downstream of a promoter, if 50% of all insertions occur in the undesired orientation. The authors should more clearly and prominently acknowledge this property of the strategy.

9) Supplemental Figure 3: the panel lettering needs to be corrected.

10) Figure 2A: I suggest highlighting the PAM in the sequence shown at the top. It should also be clarified that the sequence shown is the endogenous sequence prior to cleavage and DNA capture/integration.

11) Figure 2B: It would be helpful to show what exactly is being amplified and sequenced; the figure itself, and description in the text and Methods, is fairly confusing. The red lines in the graph itself that denote the percentage of reads could also be made more clear.

12) Lines 140-144: This sentence needs to be carefully rewritten, as in its present form, it misleadingly implies that “ORF1 and ORF2... is [sic] responsible for targeted integration” i.e. are performing the integration reaction. Being responsible for DNA excision is *not* the same as being responsible for integration, and this muddiness in writing, together with the use of “synthetic CAST,” is giving the false impression of a transposase-based insertion reaction.

13) Line 145: Delete the word “textbook” and consider replacing “seamless,” as it could be misread to suggest that these are transposition products.

14) Line 153: I’m not sure “precision” is the most appropriate word here.

15) Lines 160-178: It would be helpful if the authors add a few sentences in the Results section explaining how Insertion-Seq was performed.

13) Figure 2E:

- Are each of the chromosomes the exact same size? If the answer is no, it would make much more

sense to have the x-axis scaled according to chromosome size, rather than showing all the chromosomes equally spaced apart.

- Why are there two data points for the on-target in replicate 2 in the bottom panel? (Dark green filled circles.) In the same panel, it's confusing why the filled circles between the two replicates are not immediately above/below each other, since they correspond to the exact same coordinate on the chromosome. Having them off-center so that the lines are more visible gives the impression that these data points do not correspond to an exact chromosomal location.
- How do the authors explain the difference in the efficiencies between replicates?
- There's a problem with the use of the term "off-target"- Pong is mediating random transposition of mPing, but there is no 'target' in this case.

14) I'd encourage the authors to use a bit more nuance in the choice of "off-target" in the text and figures. As pointed out by the authors in their analyses, many (or most) of the insertion events that were profiled by Insertion-Seq are likely to be the result of Pong-only transposition, in which case the use of "off-targets" is not really relevant.

15) Lines 189-190: Have the authors actually demonstrated individual cells (or transgenic plants) that contain both knock-ins simultaneously? If this analysis was performed on a population of cells, one cannot differentiate between cells containing either one insertion or the other (or neither), versus cells containing both insertions in the same genome.

15) Figure 3: While this figure nicely summarizes a series of experiments that the authors used to extend their editing strategy to new payloads and target sites, it seems a bit too simple and lacking in rigor to be a main text figure for a Nature article. I would encourage the authors to replace panel C with more quantitative data on the relative insertion efficiencies for the variable-sized payloads and activity at new target sites.

16) Figure 3B: These data would be better represented with bar graphs, so that the difference efficiencies are not clouded by the different number of transgenic plants tested in each experiment. Currently, the figure gives the impression that the strategy using fused Cas9 performs similarly to the strategy with unfused Cas9, but in fact the insertion efficiency is 6X lower. This comparison would be more objective if the numbers were plotted as a fraction of total.

17) Please use Cas12a in place of CPF1 throughout, as this is the presently accepted nomenclature in the field.

18) Lines 278-279, 299: As mentioned above, the authors should adopt nomenclature conventions that are accepted by the rest of the field, and reserve the use of CRISPR-associated transposases (CASTs) for bona fide transposases that evolved be naturally guided by CRISPR RNAs; i.e., the phrase "synthetic CAST system" should be avoided. Reference 31 does not study/develop a CAST system (and does not name it as such).

Referee #3 (Remarks to the Author):

Liu et al. report on the set up of a novel approach enabling the CRISPR-targeted transposase-mediated insertion of foreign DNA into plant genomes.

The set up of such a technology in eukaryotes is in the focus of numerous groups adopting either natural bacterial or constructing artificial systems. To my knowledge this is the first report that the principle has been set up successfully. No question this is a major breakthrough.

The authors used an approach with a Cas9 fusion to the C-terminus of ORF2 of the rice pong TE in combination with the expression of ORF1 to achieve directed integration. Beside Cas9, Cas12a worked with the same approach, too. In Arabidopsis they were able to obtain with a one-transgene approach an insertion frequency of 6.7%, which is definitely higher than HR and NHEJ based approaches. Up to 1 kb of cargo sequences could be integrated together with mPING

So far so fantastic, but I have major concerns:

I was not able to find the integration efficiency of the constructs in figure 3C. These data have to be supplied to replace the simply ok hooks, which are by no means informative

Why does the Cas9 fusion to the C-terminus of ORF2 does not work in soybean in contrast to Arabidopsis? To use the unfused protein is for me not a serious alternative, as I am strongly worried about the high frequency of off-site integrations: Beside the targeted inversion 5 additional inversions were found in a soybean line on 4 different chromosomes. Without fusion the whole concept of targeted transposon integration by tethering has to be reevaluated. Apparently after excision mPING is preferentially inserting into existing DSBs.

Minor point: I would suggest that the author adopt to the now widely accepted nomenclature and replace Cpf1 by Cas12a. They should specify Cas12a of which organism was used.

Author Rebuttals to Initial Comments:

Referee #1 (Remarks to the Author):

In this manuscript, Liu et al. generated a synthetic CAST system by co-expressing catalytically active programmable nucleases with the active rice Pong DNA transposon system and accomplished transposase-mediated targeted insertion in the plants of Arabidopsis as well as of soybean. It is a very interesting and potentially important methodologies, which may help to accomplish targeted integration and delivery of enhancers and gene cargos.

The results are straightforward described and illuminated, however, some data need to be further confirmed or revised.

We thank Reviewer #1 for their insight and attention to detail. We have made every effort to make all of these corrections.

1. The picture used for Figure 1A-B and Supplemental Figure 1B-D are duplicated, which need to be clarified.

Images in Figure 1A and Supplemental Figure 1B (now Supplemental Figure 2) are subsets of one large microscopy experiment. The duplicated images represented the same line (ORF2-Cas9). We have revised the Supplemental Figure 1B, using a different biological replicate to represent the line of “ORF2-Cas9”. The rest of these pictures are not duplicated, and we have now mentioned in the legend of Supplemental Figure 2 that this was all part of one large experiment.

2. The authors compared the fused +ORF2-Cas9 to the unfused +ORF2, +Cas9 configuration, and claimed that their overall precision at the target site was indistinguishable. Whereas, based on the data shown in figure 2D, there are some pattern differences. If a conclusion of ‘indistinguishable’ to be drawn, statistical analysis need to be performed.

This is a good comment and we agree that there are some pattern differences in this data. We have explored this difference using statistics and added this requested statistical analysis to the discussion of Figure 2D. The unfused configuration produces slightly more seamless insertions compared to the fused configuration.

3. The authors claimed that ‘The off-target sites in the synthetic CAST lines are not directed by CRISPR/Cas9 cleavage’ and ‘these off-target sites are a product of the active mPing transposition system and not CRISPR/Cas9 cleavage’ because of ‘a very few off-target sites are in common between replicates’. The current data did not fully support the conclusion. These part need different controls to make strict comparisons.

First, as a clarification, we never saw the exact same off-target site in any of our fused or unfused configuration replicates (more replicates and other control lines have been added to this revision). Our one overlap from the previous version of this manuscript was in the same gene, but at different nucleotide locations. We have made our analysis more specific to

individual sites rather than basing it on genes. It is now clear that no off-target sites are shared between replicates.

Second, we have now added a significant amount of new data to this analysis, including several more biological replicates and the unfused version of ORF1 + ORF2 + Cas9.

Third, the fact that the off-target sites are not the same between biological replicates was only one piece of evidence that tells us that these are not CRISPR-induced off-target sites. Most importantly, there is no evidence of a gRNA target sequence at these off-target sites. We have added a new Supplemental Figure 6 that analyzes *mPing* insertion into potential off-target Cas9 cleavage positions of the genome, identified by sequence similarity to the gRNA targeting sequence. We cast a very wide net to find any potential off-target site with only 11/20 nucleotides in common with the *PDS3* gRNA that we used, and we do not find *mPing* at any of these sites.

Fourth, as a general note, whole genome sequencing is used in other recent reports to detect the off-target rate, but we believe this is not the right assay. If an off-target event occurs in a small set of cells in a tissue sample, this will be swamped out by all of the wild-type DNA at each genomic location. Assaying off-target rate by whole genome sequencing therefore significantly under-counts the true off-target rate. Our methodology of Insertion-seq is much more accurate and does not undercount the off-target rate, which is why the off-target rate of our technology seems much higher - This is due to our assay being significantly more sensitive. We have mentioned this point in the Discussion section and added Supplemental Figure 6 comparing our data if we use the methodology to map these off-target insertions used by other recent papers in this field.

4. The effect of the insertion lines either in Arabidopsis and soybean need to be illuminated, such as the gene structure and the gene expression changes, and the final phenotype.

We have added a large amount of new data that illuminates the structure and effect of the targeted insertions. First, we added the epigenetic effect of the insertions on the neighboring sequences. We found that once the *mPing* element transposes to its new location in an exon of the *ADH1* gene in Arabidopsis, it does not bring along any methylation and does not result in the *ADH1* gene being more methylated than it was before the *mPing* insertion. This data has been added as Supplemental Figure 7.

Second, in soybean, we now understand that an herbicide resistance cassette within *mPing* can drive this trait successfully. This data is in Figure 4 and Supplemental Figure 14.

Third, we have used Sanger sequencing to sequence across *mPing* targeted insertions of user-defined cargo, which is in Supplemental Figures 9, 11 and 12. For delivery of these cargos, safe-harbor locations within the genome are used, and there is no expected change on neighboring gene expression or phenotype. The major phenotype (trait) that we are

demonstrating is herbicide resistance in soybean, which is shown in Figure 4 and the new Supplemental Figure 14 (see below).

5. A case on gene functional modification to show the advantage this technology than others will clearly highlight the importance of this methodology.

We completely agree and have added this data.

We transformed soybean with a version of *mPing* that contains an herbicide resistance (BAR) cassette (promoter + ORF + terminator). We obtained plants from this transformation that have the targeted integration of this *mPing_bar* element but THESE PLANTS LACK THE ORF1+ORF2+Cas9+gRNA transgene (and donor *mPing_bar* element). Since this plant is herbicide resistant and survives selection, the BAR cassette within *mPing* must be functional.

The production of these targeted insertions in soybean is 6.7-9.8%. This represents the advantage of this system: We can perform targeted insertion of an industry-relevant trait in the crop soybean genome at a rate multiple times higher than ever previously reported. The only rates of targeted insertion that come close to this in a crop use Corteva's landing pad technology, which requires that a short sequence must first be integrated into the target site (the landing pad), followed by the insertion of the transgene (PMID 32431725). This is a longer multi-step process and requires prior sequences to be present. Our system does not need a landing pad and can be targeted to any site that CRISPR/Cas can cut (Figure 2).

Referee #2 (Remarks to the Author):

In their article, Liu et al. exploit the rice transposase, Pong [*italics are omitted throughout*], alongside the non-autonomous TE mPing, together with a programmable nuclease, Cas9 or Cas12, to achieve targeted integration of DNA payloads ~400-1,500-bp in size at a specific location in the Arabidopsis genome. After demonstrating that Pong, with or without a Cas9 fusion, is still competent for DNA excision of a genomically integrated mPing, the authors design a series of gRNAs to direct site-specific integration in Arabidopsis, as demonstrated by both qualitative PCR, quantitative plant genotyping, and high-throughput sequencing of genome-wide integration specificity. In a last set of experiments, the same strategy is used to achieve targeted insertions in the soybean genome, albeit at much less efficiency.

We thank Reviewer #2 for the large effort and attention to detail writing this review. We have made every effort to make all the changes and perform all the requested experiments.

We feel that it is important to highlight that our rate of targeted insertion in soybean from Figure 4 is 18.2% for *mPing*, which is much >5X more efficient compared to the highest rates reported previously by industry (3.4%).

Altogether, the study is well-written, clearly presented, and the experiments expertly performed; indeed, this work was a pleasure to read. Although this reviewer is not deeply familiar with plant genetics and knock-in methods, the strategy developed by the authors to co-deliver Cas9 and Pong appears to generate much higher efficiencies of targeted knock-ins than have been achieved with other methods in plants, though with the requirement that a mPing payload is first stably (and randomly) integrated into the genome, alongside the genes encoding Cas9 and Pong. A more plant-specific reviewer can better comment on the utility of this approach, as compared to other more standard (non-targeted) integration methods like use of *Agrobacterium*.

Our system uses *Agrobacterium* for every transformation (soy and Arabidopsis). We have made this more clear in the revision for non-plant biologists such as Reviewer #2.

Anytime research is discussed between fields, the known first problem to overcome is setting a common language. Many of the points below are language differences between the cell culture-based gene editing field and the field of plant biology. We have acquiesced and will use Reviewer #2's language throughout the manuscript.

From the perspective of genome engineering technology, however, the approach developed by the authors is not distinguished from many earlier strategies that already exploited Cas9 to generate DNA double-strand breaks (DSBs) that increase the rates of homology-independent targeted integration (HITI) of large DNA payloads, in a mechanism that relies on non-homologous end joining (e.g. Lackner et al., Nat Comm 2015; Schmid-Burgk et al., Nat Comm 2016; Suzuki et al., Nature 2016; Serebrenik et al., Genome Res 2019). The authors reliance and requirement for an active nuclease clearly establishes that their approach is

similarly dependent on NHEJ-based capture of DNA, and thus the published strategy also overlaps with approaches that have been published and deployed in mammalian cells using alternative transposase proteins such as piggyBac (Luo et al., Nucleic Acids Res 2017; Pallares-Masmitja et al., Nat Comm 2021; Bazaz et al., Sci Rep 2022;) or Sleeping Beauty (Kovac et al., eLife 2020). Perhaps most importantly, the authors' own data clearly and convincingly demonstrated that targeted integration works best when Cas9 is not even fused to Pong, indicating that there is almost certain to be no direct synergy or molecular interactions between the RNA-guided targeting component and the transposase-transposon components during the editing step. This conclusion, along with more general semantic considerations (see below), renders the use of cst(CRISPR-associated transposase) inappropriate to describe the present technology.

We are aware of the HITI approach and have added discussion of HITI and these references to the revised manuscript.

Typically, a crop improvement approach would not use HITI and CRISPR to excise the cargo DNA from the donor site because they could simply instead perform the 'Knock-In' approach we describe and reference in the Introduction section. In this method, researchers co-transform a linear fragment of DNA while performing CRISPR at the target site. At a low rate, the linear cargo DNA will be integrated at the target site by NHEJ, but it is very error-prone resulting in deletions of the delivered DNA and flanking insertion site.

In our manuscript we repeatedly compare the rate and benefits of our system to this currently-used Knock-In methodology because that is what the current state-of-the-art is in our field. We did not originally compare our rates and technology to other cell culture methodologies such as HITI because 1) it isn't used in the field, and 2) we felt that this would not be an apples-to-apples comparison.

However, at the Request of Reviewer #2, in this revision we have performed a new set of experiments and directly compared TATSI to HITI (Figure 3C). We find that our system in the unfused configuration is more efficient for targeted insertion, and HITI has a targeted insertion rate that is only ~55% as efficient at TATSI.

In addition, we have added Supplemental Figure 13 that demonstrates that there is no synergy between Cas9 and the transposase proteins when they are directly fused. Quite the opposite, fusion hinders the function of the transposase proteins (excision). We have clarified this point in the manuscript. The benefit of our system is not their synergy, but their sequentially ordered functions that result in the targeted insertion of DNA cargo.

We have acquiesced to Reviewer #2 and not used the term "CAST" (Crispr-Associated Transposase) to describe our technology. Instead, we have now created the new term "TATSI" (Transposase-Assisted Target Site Integration) to use throughout the revision.

Admittedly, this work is likely to be the most rigorous of all studies that have used Cas9 in conjunction with a transposase, and is the first (that I am aware of) to do so in plants.

We appreciate this positive comment from Reviewer #2.

Perhaps the most interesting mechanistic/technological value arises from the claim that their high integration efficiencies are enabled because of Pong's ability to protect the DNA ends of the excised payload prior to NHEJ capture, however this claim is not actually proven or supported with experiments. To do so, the authors could compare efficiencies of integration using Cas9 with Pong, versus using Cas9 with additional gRNAs directed to excise the mPing (or any other) payload through additional DSBs, as was developed with HITI and related methods.

The binding of transposase proteins to the ends of DNA transposons results in the excising of the element and continued binding of the ends that protects the transposon ends while the DNA is extrachromosomal. This is widely accepted in the literature (see below). The image below is from Tang et al. 2005 and shows a transposase protein binding the ends of transposable element DNA when it is extrachromosomal.

Beall EL, Rio DC. *Drosophila* P-element transposase is a novel site-specific endonuclease. *Genes Dev.* 1997 Aug 15;11(16):2137-51. doi: 10.1101/gad.11.16.2137. PMID: 9284052; PMCID: PMC316450.

Tang M, Cecconi C, Kim H, Bustamante C, Rio DC. Guanosine triphosphate acts as a cofactor to promote assembly of initial P-element transposase-DNA synaptic complexes. *Genes Dev.* 2005 Jun 15;19(12):1422-5. doi: 10.1101/gad.1317605. PMID: 15964992; PMCID: PMC1151657.

Tang M, Cecconi C, Bustamante C, Rio DC. Analysis of P element transposase protein-DNA interactions during the early stages of transposition. *J Biol Chem.* 2007 Sep 28;282(39):29002-29012. doi: 10.1074/jbc.M704106200. Epub 2007 Jul 19. PMID: 17644523.

This point has been clarified and these references have been added to the manuscript.

Reviewer #2 proposed an interesting experiment to directly compare our system to a traditional HITI experiment. We performed this experiment that Reviewer #2 suggests to compare HITI (CRISPR excised and unprotected extrachromosomal DNA) to our system (transposase excised). We tested the efficiency rate of excision and targeted integration. This experiment has been added as Figure 3C. This experiment showed that cleavage of *mPing* by CRISPR/Cas9 is efficient in our HITI experiment, but the targeted insertion is only ~55% as efficient at TATSI. We have added this analysis to the manuscript and we appreciate this comment as we think it improves our manuscript.

Altogether, given that the approach is not new, and that the mechanism of donor DNA integration is not differentiated from earlier studies, I am not sure that this will be of sufficient interest for a general audience to warrant publication in Nature.

There is no doubt that similar approaches to ours have been demonstrated in cell cultures or microbes. We never made this claim. The key advancement here is that we've moved this technology out of culture and into a global crop that badly needs this technology. What we have done is match the technology with a critical application and bridged this technical divide. We had to work hard to get this technology functional in crops.

MAJOR COMMENTS:

1) The "large, targeted DNA integration" field has gained a lot of attention in recent years, with approaches like HITI, CAST, PASTE, etc., but is at the same time suffering from a lack of detail in the way that technologies are being described and named.

This is a very good comment and is an important point. We have changed our terminology based on this and his/her other comments. We have also changed the title to reflect our improved terminology.

The results presented by the authors clearly demonstrates that *Pong* is functioning as a transposase to excise the *mPing* genetic payload -- though inactivating mutations to the *Pong* active site, and/or mutations to delete one or more TIRs from the *mPing* transposon, would be welcome additions to further prove this point.

We agree with Reviewer #2 that our data clearly demonstrates that *Pong* is a functional transposase. This has been shown multiple times in the literature. We have not made additional new mutations to ORF1 or ORF2 besides all the fusion proteins to Cas9/Cas12a. The *Pong* ORF2 protein has a DDE motif that is well-conserved in transposase proteins and it has been shown many times in the literature that mutating this site will result in a lack of excision. We also did not mutate the *mPing* TIRs, as again, we know this will result in a lack of excision. Excision is critical in our system, and we did not make mutations that we know will abolish excision.

However, we did add Supplemental Figure 13 in which we study fusions to the *Pong* TE ORF2 protein on the excision of *mPing* in yeast.

However, it is clear that DNA integration is NOT occurring through a transposase-based mechanism, but is instead resulting from NHEJ-based capture of the linearized and excised DNA. This can be concluded from the prior literature, from the authors' experimental results using dCas9, and from the presence of hallmark indel mutations in the integration products.

We agree that the transposase is not mediating insertion, and this is further clarified in our model in Figure 2. We understand that the integration (and specifically the determination of the

insertion site) is not occurring in the same way as a bonafide transposition event, and have clarified this point in the revision. We know that NHEJ is important (as it is in nearly all plant DNA repair and gene edits) in this insertion process and have clarified this point in the Discussion section. We have also removed all claims of “transposition” in our system.

Thus, it is inappropriate to refer to this technology as a “synthetic CAST,” and all references to this should be removed from the paper. Mis-use of this term will confuse readers, giving the impression that a transposase is generating the DNA integration product, and muddying the field.

We see Reviewer #2's point and have removed the term “CAST” from the description of our technology. Instead we will use our new term TATSI (Transposase-Assisted Target Site Integration).

Separately — even if Pong were fused to dCas9 and being repurposed for the DNA integration reaction — it would still be inappropriate to repurpose the term CAST. CRISPR-associated transposases (CASTs) refer to specific Tn7-like transposons that have naturally coopted nuclease-deficient CRISPR-Cas systems for RNA-guided transposition. If everything involving a CRISPR protein and a transposase becomes a “CAST,” then the meaning and significance of this well-defined term will be sufficiently eroded so as to essentially render it unusable for the field where CASTs are currently and appropriately used.

We have agreed with Reviewer #2 not to use the term “CAST” to describe our technology.

2) Relating to #1, the use and labeling of TSD (target-site duplication) in Figure 1D, and throughout the rest of the figures, is misleading and should be removed. TSDs are hallmark features that occur upon transposon integration, and they refer to duplications that arise from gap fill-in upon resolution of the transposition product. Here, the authors are mis-using the term to refer to a portion of the flanking sequence that derives *from the original donor site* (where it was a TSD for that first insertion).

Reviewer #2 is generally correct that TSDs occur at the insertion site.

This point has been studied for the *mPing/Pong* transposon system. A previous publication (referenced below) showed that for *mPing/Pong*, a repeated TSD-like sequence at the DONOR site is essential for transposition. This is unusual for a transposable element, but our author list contains the world's expert on the *mPing* transposable elements (Nathan Hancock) who has studied this system extensively.

At Reviewer #2's request, we do not call these TSDs, but rather have changed our terminology to “flanking TTA/TAA sites”.

When we build our custom donor sites for *mPing* cargo, we add these flanking TTA/TAA sites to the donor site. We have clarified this point throughout the manuscript to avoid this confusion.

As an interesting side note, since the flanking TTA/TAA sites are required for efficient excision from the donor site (see publication below), nearly all of our TATSI insertions (that fail to have

perfect seamless insertions on either side) will not be excisable. This is a benefit to our system because it results in the fact that cargo delivered to the targeted site will not be able to excise back out.

We have clarified each of these points in the revision.

From the paper below:

“The TSDs of both elements [Tourist (mPing) and Stowaway] play a role in element excision, but only the mPing TSDs actively participate in excision site repair. Our data suggests that Tourist-like elements excise with staggered cleavage of the TSDs, which provides microhomology that facilitates precise repair.”

Gilbert DM, Bridges MC, Strother AE, Burckhalter CE, Burnette JM 3rd, Hancock CN. Precise repair of mPing excision sites is facilitated by target site duplication derived microhomology. *Mob DNA*. 2015 Sep 7;6:15. doi: 10.1186/s13100-015-0046-4. PMID: 26347803; PMCID: PMC4561436.

A TSD is clearly not being generated at the new targeted insertion site, because (a) as established by the authors, these are not even transposition products; and (b) the TSD is rarely duplicated anyways, since their data in Figure 2 show that it is frequently mutated during the NHEJ-based capture step

We agree and have shown this point in Figure 2.

Although I understand that transposon-flanking sequences are excised alongside the mPing element, as part of the excision reaction — similar to other elements like piggyBac — the authors should modify their terminology to avoid confusing readers into thinking that these are transposition products. I suggest labeling this as “flanking TTA” or simply “TTA,” and then clarifying better in the text what the origin of this sequence is. (For a general, non-specialist journal, the authors would help out readers by also including a supplementary figure outlining the known chemical steps and sequence features of canonical Pong/mPing transposition, including the way that mPing transposable elements are excised at the TIR boundaries, together with flanking TTA sequences, presumably as 5' overhangs.)

We have made the requested changes. We have labeled the TTA at the donor site as “flanking TTA”.

Because our system does not make perfect TSDs, we only use the term “TSD” to describe a perfect free transposition event to compare and contrast our system to.

As described above, we have removed any claims of ‘true transposition’ by our system.

We have also included a new Supplemental Figure 1 on the background of what is known in the literature about *mPing/Pong* transposition mechanisms.

3) As mentioned above, it would be an interesting and exciting finding, to prove that protection of the donor DNA ends from cellular degradation is a source for the higher integration efficiency

in this study, relative to earlier approaches. However, there is no explicit experimental support for statements suggesting that full-length insertions result from “binding of the TE by the transposase proteins while extra-chromosomal, protecting the DNA ends from nucleases and deletions” (line 134-136). I suggest an experiment in which Pong and Cas9 are expressed separately from an mPing derivative that is flanked by two Cas9 cut sites, such that Cas9 expression with three gRNAs will both cleave the target site and excise the mPing payload, similar to HITI. Then, in experiments tested with or without catalytically inactive Pong, the role of Pong in stimulating integration efficiency by protecting the ends – when mPing excision itself is due to Cas9 cleavage activity – can be directly tested and quantified. Importantly, these results would be quite informative for targeted DNA integration applications in other cellular contexts, including for HITI-like applications in mammalian cells, if demonstrated that DNA protection by transposase (or other proteins) increases both DNA integration efficiency and product purity.

We have taken Reviewer #2’s suggestion seriously and have executed the HITI portion of this proposed experiment. This data is now integrated into Figure 3C as described above.

Further, we want to comment that transposase proteins are known to bind the TIR ends of transposable elements while the element is extrachromosomal. This is widely accepted and I’ve copied the references from above here. The image shows the transposase proteins binding to the ends of the extrachromosomal transposable element and creating a loop of the transposable element internal sequence. The key question is whether this binding to the ends is PROTECTING the ends. We took this for granted, because we assumed this would be a logical extension of the binding. However, Reviewer #2 brings up a key point regarding if binding also leads to protection and enhanced target site integration.

Tang M, Cecconi C, Kim H, Bustamante C, Rio DC. Guanosine triphosphate acts as a cofactor to promote assembly of initial P-element transposase-DNA synaptic complexes. *Genes Dev.* 2005 Jun 15;19(12):1422-5. doi: 10.1101/gad.1317605. PMID: 15964992; PMCID: PMC1151657.

Based on this comment by Reviewer #2, we have removed the direct claims that the transposase protein protects the free ends of the transposable element DNA when it is extrachromosomal. However, since our TATSI system works at a higher efficiency than HITI (data in Figure 3C), we speculate that this is one reason why TATSI is more efficient. We clearly mark this as speculation in the manuscript.

MINOR COMMENTS

1) Please add ladder sizes and expected sizes to all electrophoresis gels and Western blot gels throughout the paper.

We have added these sizes throughout the manuscript.

2) Chromatograms of Sanger sequencing results should be shadowed differently (e.g. in Fig. 1D), so that readers can visualize which regions of the insets correspond to the endogenous PDS3 sequence, the mPing TIR, or the flanking sequence derived from the donor site (i.e. the TTA).

This is a good idea and we have adjusted all of the sequencing images to denote which bases come from what regions of *mPing*, the flanking TTA/TAA and the target site.

3) The introduction section would be improved by including more general details about the Pong transposase and general Pong/mPing transposition mechanism.

We added this detail as well as a new Supplemental Figure 1 requested by Reviewer #2 that details what is known from the literature about *mPing* transposition.

4) The introduction is missing a number of key citations to earlier studies that developed similar approaches to exploit NHEJ-based capture of linear DNA at sites of Cas9-based DNA cleavage. (E.g. Lackner et al., Nat Comm 2015; Schmid-Burgk et al., Nat Comm 2016; Suzuki et al., Nature 2016; Serebrenik et al., Genome Res 2019)

The discussion of these papers and their references have been added to the revision.

5) Lines 56-60: There are many Class II DNA transposons that do not mobilize through a cut-and-paste mechanism; please update the text accordingly.

We have updated the text accordingly.

More importantly, the statement that “TE-encoded transposase proteins... cleave the genome at the integration site” is inaccurate. The DNA is not cleaved as in a hydrolysis mechanism, but is instead subjected to a transesterification reaction, or strand-transfer reaction. This is distinct from what the word “cleavage” implies. (Similar comment, relating to the use of the word “cut” in line 66.)

We have improved our language and we now do not use the general terms “cleavage” and “cut” with the transposase.

6) Figure 1B,C: These panels are difficult to follow without a schematic of the assay provided directly in the same figure. I suggest moving the corresponding panels from the supplement into the main text figure, so that a reader view all the information together.

We have moved the cartoon descriptions of primer placement into Figure 1 as panels B and C.

7) I find the schematic in Fig. 1E to be misleading, related to major comments #1 and #2 above. When viewed in isolation, and/or together with the name “synthetic CAST,” this gives the impression that the Pong transposase proteins are performing the targeted insertion reaction, rather than two completely separate molecular events: DNA DSB formation by Cas9, and NHEJ-based capture of the linear DNA excised by Pong. Fig. 1E and 2C should be combined into a single figure that clearly delineates the likeliest biochemical pathway.

We have agreed not to use the term “synthetic CAST” to avoid confusion.

We have added detail to Figure 1F so it is clear to the reader that the transposase does not mediate the insertion nor protect the transposon DNA when it is extrachromosomal. However, we have not combined these models into one, as they show different aspects of the system: Figure 1F shows that TATSI is possible and is a summary of what was discovered in Figure 1. Figure 2 is our determination of the specific mechanisms at play, and this model then shows the role of NHEJ.

We favor keeping these two panels as separate schematics that help the reader understand the separate results of Figure 1 and Figure 2. If we combine them, there will be a lot of detail in Figure 1 that the reader won't understand until later in the paper, and vice-versa. Instead, Figure 1 shows the overall scheme with excision and insertion, while Figure 2 adds detail to the understanding of what is occurring at the insertion site.

8) The figure schematics and Results/Discussion nearly completely overlook the fact that insertions will be bidirectional using the approach developed by the authors. This will be particularly problematic for experiments in which a gene-of-interest is e.g. integrated downstream of a promoter, if 50% of all insertions occur in the undesired orientation. The authors should more clearly and prominently acknowledge this property of the strategy.

We now clearly state this point in a new prominent sentence in the Discussion section.

9) Supplemental Figure 3: the panel lettering needs to be corrected.

We have fixed this oversight made in the legend to Supplemental Figure 3.

10) Figure 2A: I suggest highlighting the PAM in the sequence shown at the top. It should also be clarified that the sequence shown is the endogenous sequence prior to cleavage and DNA capture/integration.

This is a good idea and we have added the requested annotation to Figure 2A.

11) Figure 2B: It would be helpful to show what exactly is being amplified and sequenced; the figure itself, and description in the text and Methods, is fairly confusing. The red lines in the graph itself that denote the percentage of reads could also be made more clear.

We have added the placement of the PCR primers to the cartoon of this assay into Figure 2B. We have clarified the description of this experiment in the text and Methods section. The red lines can only be as thick as the space of that specific nucleotide on the X-axis, so we have labeled the bar values so it is more clear to the reader what percent of the reads this bar represents.

12) Lines 140-144: This sentence needs to be carefully rewritten, as in its present form, it misleadingly implies that “ORF1 and ORF2... is [sic] responsible for targeted integration” i.e. are performing the integration reaction. Being responsible for DNA excision is *not* the same as being responsible for integration, and this muddiness in writing, together with the use of “synthetic CAST,” is giving the false impression of a transposase-based insertion reaction.

We have agreed not to use the term “synthetic CAST” to avoid confusion.

We have changed our writing to be more accurate about the unknown role of the transposase proteins during integration. The sentence pointed out by Reviewer #2 has been corrected in the manuscript.

13) Line 145: Delete the word “textbook” and consider replacing “seamless,” as it could be misread to suggest that these are transposition products.

We have changed our writing to be more accurate. These terms are common in plant genome engineering but we have changed them to be more clear and accurate.

14) Line 153: I’m not sure “precision” is the most appropriate word here.

We have changed our writing to be more accurate.

15) Lines 160-178: It would be helpful if the authors add a few sentences in the Results section explaining how Insertion-Seq was performed.

We have added this detail on Insertion-Seq to the Results section.

13) Figure 2E:

- Are each of the chromosomes the exact same size? If the answer is no, it would make much more sense to have the x-axis scaled according to chromosome size, rather than showing all the chromosomes equally spaced apart.

This is a good point. We have scaled X-axis for soy and Arabidopsis chromosomes for the revision. We have added several more lines and replicates of Insertion-seq and our display of this data is improved.

- Why are there two data points for the on-target in replicate 2 in the bottom panel? (Dark green filled circles.) In the same panel, it's confusing why the filled circles between the two replicates are not immediately above/below each other, since they correspond to the exact same coordinate on the chromosome. Having them off-center so that the lines are more visible gives the impression that these data points do not correspond to an exact chromosomal location.

We had attempted to make this easier to understand by offsetting the points, but clearly this was our mistake. In addition, the filled circle on the X-axis is our oversight and appears as a second on-target site. These fixes have been made for the revision.

- How do the authors explain the difference in the efficiencies between replicates?

There isn't as much difference between the replicates as the original figure seems to show. Biological replicate 2 simply generated a lot more *mPing*-containing sequenced reads (both on-target and free transpositions) compared to biological replicate 1. You can see this much better in the Y-axis of the two replicates now that we have split them out and show them separately. Therefore, the relative enrichment of on-target vs. free transpositions is very similar between replicates. We have added the comparison between replicates as Supplemental Figure 6A. We adjusted this data display to clarify this point in the revision.

- There's a problem with the use of the term "off-target"- Pong is mediating random transposition of *mPing*, but there is no 'target' in this case.

We have altered this term "off-target" to "free transposition of *mPing*".

14) I'd encourage the authors to use a bit more nuance in the choice of "off-target" in the text and figures. As pointed out by the authors in their analyses, many (or most) of the insertion events that were profiled by Insertion-Seq are likely to be the result of Pong-only transposition, in which case the use of "off-targets" is not really relevant.

We have changed the term "off-target" to "free transposition of *mPing*".

We do want to make a point here about "off-targets". Most of the field detects off-targets based on whole genome sequencing. This isn't a good method because if the off-targets are different in different cell populations within the sample, they get washed out by the abundance of WT DNA at each locus and not reported in the final result. Also, recent papers in this field have only investigated certain sites for off-target insertions. Our method of Insertion-seq is much more sensitive and accurately reports off-targets / free transpositions, while other recent papers use inferior methodology to skirt around this problem and report very few off-targets. We have added Supplemental Figure 6B to make a direct comparison for the reader.

15) Lines 189-190: Have the authors actually demonstrated individual cells (or transgenic plants) that contain both knock-ins simultaneously? If this analysis was performed on a population of cells, one cannot differentiate between cells containing either one insertion or the other (or neither), versus cells containing both insertions in the same genome.

In the experiment with two targeted insertion sites, we have no evidence that these insertions are in the same cell. We can't be sure, which is why we were careful not to make this claim, but rather claim that only the plant (not single cell) has both targeted insertions. We have clarified this point in the manuscript text.

15) Figure 3: While this figure nicely summarizes a series of experiments that the authors used to extend their editing strategy to new payloads and target sites, it seems a bit too simple and lacking in rigor to be a main text figure for a Nature article. I would encourage the authors to replace panel C with more quantitative data on the relative insertion efficiencies for the variable-sized payloads and activity at new target sites.

This is a very good idea and we performed the plant-by-plant quantification of excision and insertion rates to add as Figure 3C.

16) Figure 3B: These data would be better represented with bar graphs, so that the difference efficiencies are not clouded by the different number of transgenic plants tested in each experiment. Currently, the figure gives the impression that the strategy using fused Cas9 performs similarly to the strategy with unfused Cas9, but in fact the insertion efficiency is 6X lower. This comparison would be more objective if the numbers were plotted as a fraction of total.

We originally had this data as a bar graph, but changed it to the current form to be able to show that targeted integration occurs for only a subset of the plants that have transposable element excision. This 'subset' aspect was going to be difficult to easily demonstrate with bar graphs. Therefore, we have now included both data displays. We added a bar graph of this data in response to this comment by Reviewer #2. This bar graph of the data is now Figure 3C.

17) Please use Cas12a in place of CPF1 throughout, as this is the presently accepted nomenclature in the field.

We have made this change to the revised manuscript.

18) Lines 278-279, 299: As mentioned above, the authors should adopt nomenclature conventions that are accepted by the rest of the field, and reserve the use of CRISPR-associated transposases (CASTs) for bona fide transposases that evolved be naturally guided by CRISPR RNAs; i.e., the phrase "synthetic CAST system" should be avoided. Reference 31 does not study/develop a CAST system (and does not name it as such).

As discussed above, we will not use the term CAST.
We changed our referencing for the above mentioned lines to be more accurate.

Referee #3 (Remarks to the Author):

Liu et al. report on the set up of a novel approach enabling the CRISPR-targeted transposase-mediated insertion of foreign DNA into plant genomes.

The set up of such a technology in eukaryotes is in the focus of numerous groups adopting either natural bacterial or constructing artificial systems. To my knowledge this is the first report that the principle has been set up successfully. No question this is a major breakthrough.

We appreciate Reviewer #3 viewing our work as a major breakthrough. The comments Reviewer #3 provided were very helpful for improving the manuscript.

The authors used an approach with a Cas9 fusion to the C-terminus of ORF2 of the rice pong TE in combination with the expression of ORF1 to achieve directed integration. Beside Cas9, Cas12a worked with the same approach, too. In Arabidopsis they were able to obtain with a one-transgene approach an insertion frequency of 6.7%, which is definitely higher than HR and NHEJ based approaches. Up to 1 kb of cargo sequences could be integrated together with mPING

So far so fantastic, but I have major concerns:

I was not able to find the integration efficiency of the constructs in figure 3C. These data have to be supplied to replace the simply ok hooks, which are by no means informative

This is a good suggestion also made by Reviewer #2, and we have now performed and added this plant-by-plant experiment to measure the rate of excision and targeted insertion of each of the different cargos in Figure 3D. We have added the rates of excision and targeted insertion to Figure 3C in the revised manuscript.

Why does the Cas9 fusion to the C-terminus of ORF2 does not work in soybean in contrast to Arabidopsis? To use the unfused protein is for me not a serious alternative, as I am strongly worried about the high frequency of off-site integrations: Beside the targeted inversion 5 additional inversions were found in a soybean line on 4 different chromosomes. Without fusion the whole concept of targeted transposon integration by tethering has to be reevaluated. Apparently after excision mPING is preferentially inserting into existing DSBs.

Reviewer #3 is correct that the previous linkage of these two proteins did not function in soybean as it does in Arabidopsis. To address the question of why, in Figure 4 we show that the transposase in the soybean fusion protein functions (albeit at a reduced rate) but the Cas9 activity is destroyed. We know that this same version of Cas9 (unfused) is functional in soybean, so therefore the issue is with the protein-protein linkage. In this revision, we have now performed experiments with a longer linker to fuse ORF2 and Cas9. We have now generated an ORF2-Cas9 fused transposase system that can perform targeted insertions at a high rate (16%).

We disagree with Reviewer #3 that not having a functional fusion causes the utility of this system to be reevaluated. As we describe, off-target insertions happen with all targeted insertion technology. Other papers use whole genome sequencing to minimize the detection of these off-target sites, since they are diluted out in a pool of cells that have different off-target insertions. Instead, using our approach to amplify the ends of the inserted DNA is significantly more sensitive and is the proper way to measure off-site insertions. Also, other recent papers in the field produce whole-genome datasets but only interrogate them for likely off-target insertion sites. We replicated this analysis in the new Supplemental Figure 6B and if we use their same criteria, we also find a low off-target rate. Furthermore, in crops these off-target insertions do not represent a major concern for product development because of the trait introgression that will be performed with the targeted insertion event. All traits go through this introgression because the crop germplasm is constantly changing and evolving year after year. Of course we are working to reduce the off-target rate, but generating an efficiently-working tool for on-target delivery of cargo is the primary concern of the field and major benefit of our current system.

We also disagree with Reviewer #3 that *mPing* is inserted into existing DSBs. There is no evidence that the off-target / free transposition sites are existing DSBs. Rather, the random transposition sites match the known TTA/TAA insertion site preference of a freely transposing *mPing* (see reference below). This point was also missed by Reviewer #1, so we have improved our writing and clarity of this point.

Yang G, Zhang F, Hancock CN, Wessler SR. Transposition of the rice miniature inverted repeat transposable element *mPing* in *Arabidopsis thaliana*. *Proc Natl Acad Sci U S A*. 2007 Jun 26;104(26):10962-7. doi: 10.1073/pnas.0702080104. Epub 2007 Jun 19. PMID: 17578919; PMCID: PMC1904124.

Minor point: I would suggest that the author adopt to the now widely accepted nomenclature and replace Cpf1 by Cas12a. They should specify Cas12a of which organism was used.

We have made these changes to the revised manuscript. The organism is specified in the Methods section.

Reviewer Reports on the First Revision:

Referees' comments:

Referee #1 (Remarks to the Author):

Thank the authors. All my questions were addressed. I do not have further comments.

Referee #2 (Remarks to the Author):

Since the first version of their paper, Liu et al. made significant strides to implement my suggested changes, from minor comments, such as making the language of the text more accessible to non-biologists, to more major concerns that were expressed in the initial review. One major concern was the comparison of TATSI to HITI, which the authors now include in the revised manuscript. I am also appreciative that the authors considered carefully my recommendation to avoid use of the term CAST. The overall language and presentation of data in the revised manuscript is much improved, and more accurately reflects the data and technology described in the study.

There are still some limitations in the data presentation that could be improved, which are outlined below.

COMMENTS:

1. Figure 1F: please add an explicit DNA cleavage step in the right part of this schematic. This is critical for a transparent and accurate representation of the key molecular steps in TATSI.
2. Also relating to Figure 1F, and throughout: I find the triangular representation of the mPing DNA to be quite confusing and unconventional (though I don't know the mPing field). I strongly suggest that the authors adopt a more traditional schematic for representing this double-stranded DNA species.
3. Figure 2E: The new representation of genome-wide integration data in the revised manuscript is borderline unintelligible and does a poor job of communicating these results. The radial depiction of these data renders comparisons between different experiments or replicates prohibitively challenging, and makes it very difficult to differentiate axis lines from data points. Other visualization choices are difficult to rationalize, such as a grey line that obscures the most important on-target data point, and showing the on-target in red nearby all the chromosome 5 data that are shown in a very similar color. This entire figure panel needs to be carefully reimagined and remade, to facilitate better accessibility by readers.
4. Figure 3B: The labels seem to be mixed up, relative to Figure 3C. Also, why have the numbers changed from the previous version (6.7% versus 6.5%)? Are these new data, or were they analyzed differently than in the original submission?

5. Figure 2A: The authors underlined the PAM, as was suggested in the previous review, but now the sequence at the top has no relationship to the graph itself, because the x-axis was inexplicably expanded by hundreds of base pairs, relative to the original version. What was the thinking behind this alteration?

6. Figure 2B: I still don't understand the way these data are shown. Why is there a peak at "0-bp"? How should the reader interpret this? I find the overall figure confusing, and this was also the case in my original reading of the manuscript.

7. Figure 3D: "Targeted" is misspelled.

8. Figure 3: the checkboxes should be removed. Simply show the schematics together with the bar graph, but without the green checkboxes. The genome editing field is sufficiently advanced so as to allow authors to dispense with panels that present binary "yes/no it worked" information.

Referee #3 (Remarks to the Author):

I read the revised version and the rebuttal letter of Liu et al with interest. The authors included a wealth of new data in the resubmitted version, which definitely addresses many concerns raised of all reviewers. This is a very important contribution interesting for a wide audience

I was happy to see the quantification of excision and insertion frequencies depending on the cargo sizes (figure 3). Moreover, the newly added demonstration of a functional Cas9 fusion in soybean is important, as it shows similar efficient integration efficiency than the trans approach - although excision and mutation induction is less efficient. For me this indicates that it might well be worthwhile to optimize enzyme activities within the fusion. If the number of excised copies and the DSB in the target could be enhanced, one should expect even more site-specific but less random insertions. It might be worth to discuss this point also in relation to Cas12a: Did the authors use LbCas12a or As Cas12a? As the efficiency of LbCas12a was recently dramatically improved for plant use by the Puchta and Qi groups it might be worth using it in future for TATSI.

Author Rebuttals to First Revision:

Referees' comments:

Referee #1 (Remarks to the Author):

Thank the authors. All my questions were addressed. I do not have further comments.

Referee #2 (Remarks to the Author):

Since the first version of their paper, Liu et al. made significant strides to implement my suggested changes, from minor comments, such as making the language of the text more accessible to non-biologists, to more major concerns that were expressed in the initial review.

One major concern was the comparison of TATSI to HITI, which the authors now include the revised manuscript. I am also appreciative that the authors considered carefully my recommendation to avoid use of the term CAST. The overall language and presentation of data in the revised manuscript is much improved, and more accurately reflects the data and technology described in the study.

We thank Reviewer #2 for their attention to detail and for appreciating our efforts.

There are still some limitations in the data presentation that could be improved, which are outlined below.

COMMENTS:

1. Figure 1F: please add an explicit DNA cleavage step in the right part of this schematic. This is critical for a transparent and accurate representation of the key molecular steps in TATSI.

We have added a "Cleavage" label to this figure as requested.

2. Also relating to Figure 1F, and throughout: I find the triangular representation of the mPing DNA to be quite confusing and unconventional (though I don't know the mPing field). I strongly suggest that the authors adopt a more traditional schematic for representing this double-stranded DNA species.

Based on this comment, we made this change throughout all the Figures. Representing a TE as a triangle to denote where it is inserted into a specific location in a genome is commonplace in the field (see the figure attached below from this recent *Nature* publication: PMID 35948641). However, when the TE is extrachromosomal (not integrated), we have changed the display to represent the TE as a linear fragment of DNA. We have re-drawn *mPing* throughout the Figures for consistency, including a major redrawing of the same model in Extended Data Figure 1.

Fig. 4: Lx9c11-RegoS rescues phenotypic changes in the immune response to CVB4 in *Lx9c11*^{-/-} mice.

3. Figure 2E: The new representation of genome-wide integration data in the revised manuscript is borderline unintelligible and does a poor job of communicating these results. The radial depiction of these data renders comparisons between different experiments or replicates prohibitively challenging, and makes it very difficult to differentiate axis lines from data points. Other visualization choices are difficult to rationalize, such as a grey line that obscures the most important on-target data point, and showing the on-target in red nearby all the chromosome 5 data that are shown in a very similar color. This entire figure panel needs to be carefully reimagined and remade, to facilitate better accessibility by readers.

Based on this comment, we have removed all radial depiction of insertion sites from the manuscript. We have reverted back to visualizing the insertion sites as linear plots similar to 'Manhattan plots'. This includes Figure 2, Figure 4 and Supplemental Figure 6. We have removed the different colors for each chromosome and highlighted the on-target data point to make it easier to see.

4. Figure 3B: The labels seem to be mixed up, relative to Figure 3C. Also, why have the numbers changed from the previous version (6.7% versus 6.5%)? Are these new data, or were they analyzed differently than in the original submission?

We have fixed the typo in Figure 3B in which we mixed-up the labels.

We have double-checked the rate of targeted insertion with the ORF2-Cas9 fused protein and it is correct at 6.7%.

5. Figure 2A: The authors underlined the PAM, as was suggested in the previous review, but now the sequence at the top has no relationship to the graph itself, because the x-axis was inexplicably expanded by hundreds of base pairs, relative to the original version. What was the thinking behind this alteration?

We were attempting to show how precise the integration site is on a more broad scale. Based on Reviewer #2's comments, we have reverted this scale back to the level that corresponds to the above sequence, as it was in the first version of the manuscript.

6. Figure 2B: I still don't understand the way these data are shown. Why is there a peak at "0-bp"? How should the reader interpret this? I find the overall figure confusing, and this was also the case in my original reading of the manuscript.

We have improved the x-axis label on Figure 2B. We have also improved the figure legend to be more clear about the x-axis means.

7. Figure 3D: "Targeted" is misspelled.

We have fixed this typo. Due to comment #8 below this label was completely removed from the figure.

8. Figure 3: the checkboxes should be removed. Simply show the schematics together with the bar graph, but without the green checkboxes. The genome editing field is sufficiently advanced so as to allow authors to dispense with panels that present binary "yes/no it worked" information.

We have removed the checkboxes. We have also added color code consistent with the color of the data bars in panel C.

Referee #3 (Remarks to the Author):

I read the revised version and the rebuttal letter of Liu et al with interest. The authors included a wealth of new data in the resubmitted version, which definitely addresses many concerns raised of all reviewers. This is a very important contribution interesting for a wide audience.

We thank Reviewer #3 for their interest and comments below.

I was happy to see the quantification of excision and insertion frequencies depending on the cargo sizes (figure 3). Moreover, the newly added demonstration of a functional Cas9 fusion in soybean is important, as it shows similar efficient integration efficiency than the trans approach - although excision and mutation induction is less efficient. For me this indicates that it might well be worthwhile to optimize enzyme activities within the fusion. If the number of excised copies and the DSB in the target could be enhanced, one should expect even more site-specific but less random insertions.

Reviewer #3 is correct that fusing ORF2 to Cas9 reduces the efficiency of both proteins (this is shown in Figure 3B, Figure 3C and Extended Data Figure 13). We agree with Reviewer #3 that optimizing and improving the fusion protein to increase the number of excised *mPing* copies and on-target DSB breaks is important and is a major future direction of ours.

It might be worth to discuss this point also in relation to Cas12a: Did the authors use LbCas12a or As Cas12a? As the efficiency of LbCas12a was recently dramatically improved for plant use by the Puchta and Qi groups it might be worth using it in future for TATSI.

We used LbCas12a, and this point has been clarified in the Results section, Methods and Supplemental Figure Legend.

We thank Reviewer #3 for pointing out the recent improvements to LbCas12a and we hope to test these improved versions in the future.

Reviewer Reports on the Second Revision:

Referees' comments:

Referee #2 (Remarks to the Author):

I am mostly pleased with the revised manuscript.

A few lingering comments:

1) Figure 2b is still confusing to me. Specifically, the legend states "The percent of reads indicating how much of the mPing TE was delivered to the targeted insertion site. The x-axis is the position (base site) along the 430 bp mPing element." So what is the meaning of a peak at coordinate 0 bp? Is "0 bp" how much of the mPing TE was delivered to the targeted insertion site? This is still confusing and unclear, both in the figure panel itself and the legend.

2) Figure 2e: Thank you, this is much better than before. However, I suggest a different nomenclature for "+ORF1,+ORF2-Cas9" since the fusion (ORF-Cas9) will easily be confused with "-Cas9" meaning that Cas9 was omitted.

3) Figure 3d should precede Figure 3c, so that the cargo architectures are presented/displayed before showing data with them.

Author Rebuttals to Second Revision:

Referees' comments:

Referee #2 (Remarks to the Author):

I am mostly pleased with the revised manuscript.

We thank Reviewer #2 for all of their help with our manuscript.

A few lingering comments:

1) Figure 2b is still confusing to me. Specifically, the legend states "The percent of reads indicating how much of the mPing TE was delivered to the targeted insertion site. The x-axis is the position (base site) along the 430 bp mPing element." So what is the meaning of a peak at coordinate 0 bp? Is "0 bp" how much of the mPing TE was delivered to the targeted insertion site? This is still confusing and unclear, both in the figure panel itself and the legend.

The 0 bp indicates that all of *mPing*, all the way to the edge base pair, was delivered to the target site.

We have changed the X-axis label and figure legend to make this point as clear as possible.

2) Figure 2e: Thank you, this is much better than before. However, I suggest a different nomenclature for "+ORF1,+ORF2-Cas9" since the fusion (ORF-Cas9) will easily be confused with "-Cas9" meaning that Cas9 was omitted.

Based on this comment, we have changed the label "+ORF2,-Cas9" to "+ORF2" throughout the entire manuscript and figures.

3) Figure 3d should precede Figure 3c, so that the cargo architectures are presented/displayed before showing data with them.

We tried to make this change, but the current 3c has quantitative data that relates to panel 3b. We discuss 3b together with 3c in the manuscript before introducing the other experiments from 3c and 3d. Therefore we couldn't make this swap and still reference the panels in order, and did not make this requested change.